# `AutoSkills`: Automatically Constructing Skill Knowledge Bases for Agents

## Abstract

Learning from experience is critical for building capable large language model (LLM) agents, yet prevailing self-evolving paradigms remain inefficient: agents learn in isolation, repeatedly rediscover similar behaviors from limited experience, resulting in redundant exploration and poor generalization. To address this problem, we propose `AutoSkills`, a fully automated framework for constructing a **plug-and-play skill knowledge base** that can be reused across agents and environments. `AutoSkills` operates through a fully automated pipeline built on three synergistic innovations: *(i) Multi-Level Skills Design*, which distills raw trajectories into three-tiered hierarchy of strategic plans, functional skills, and atomic skills; *(ii) Iterative Skills Refinement*, which automatically revises skills based on execution feedback to continuously improve library quality; and *(iii) Exploratory Skills Expansion*, which proactively generates and validates novel skills to expand coverage beyond seed training data. Using a strong backbone agent (GLM-4.6), we automatically build a reusable skill library and evaluate its transferability on challenging long-horizon, user-interactive benchmarks, including AppWorld, BFCL-v3, and $\tau^2$-Bench. Experiments show that AutoSkills consistently improves task success and execution efficiency when plugged into weaker base agents, highlighting the importance of structured, hierarchical experience representations for generalizable agent learning.

## 1. Introduction

Large language model (LLM) based agents (OpenAI, 2025; DeepSeek-AI, 2025; Team et al., 2025b; Yang et al., 2025) have recently demonstrated remarkable progress in long-

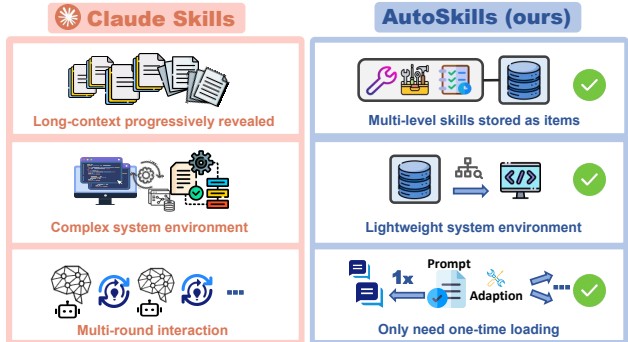

*Figure 1.* Claude Skills follow a long-context, progressively disclosed format, which requires a complex sandboxing system and multiple interactions, thereby posing challenges to robust reasoning. In contrast, AutoSkills adopts a hierarchical, itemized representation that can be stored and retrieved via a lightweight retrieval module and injected into the system prompt in one time, making it easier to transfer across base models.

horizon decision making with tools, enabling complex behaviors such as API calling (Trivedi et al., 2024; Patil et al., 2025; Li et al., 2025), web navigation, (Yao et al., 2023; Zhou et al., 2024; Mialon et al., 2023), scientific discovery (Ou et al., 2025; Liu et al., 2025; Qiao et al., 2025; Novikov et al., 2025), and interactive assistants (Barres et al., 2025; Yao et al., 2024; He et al., 2025). Despite these advances, most agents still approach each new task largely *from scratch*, relying on direct reasoning or limited task-specific demonstrations. This paradigm is costly, brittle, and fundamentally at odds with how intelligent systems are expected to accumulate and reuse experience over time.

A natural resolution is to enable agents to ***learn from experience*** (Sutton, 2025). Recent work has explored self-evolving agents that iteratively reflect on past executions and improve their behavior over time (Wang et al., 2025c; Fang et al., 2025b; Zhao et al., 2024; Xu et al., 2025; Cao et al., 2025). While promising, these approaches often fail to deliver scalable and transferable gains. In practice, experience learning typically suffers from three structural limitations. *(1) Isolated learning*: agents execute the same tasks repeatedly and re-extract similar experiences independently, leading to substantial redundancy. *(2) Weak Generalization of Experience*: in complex environments, high-quality training data are scarce, so the mined experiences often

[1]Anonymous Institution, Anonymous City, Anonymous Region, Anonymous Country. Correspondence to: Anonymous Author <anon.email@domain.com>.

Preliminary work. Under review by the International Conference on Machine Learning (ICML). Do not distribute.

transfer poorly to novel tasks. *(3) Model Capability Bottleneck*: when experience is harvested solely through an agent's own exploration and reflection, what can be extracted is ultimately capped by the agent's current capability frontier. These challenges point to a more fundamental question: **What form of experience can be broadly reusable across agents of varying capabilities and across diverse environments?**

Existing work has proposed multiple representations of experience, such as insights (Cao et al., 2025; Ouyang et al., 2025), workflows (Wang et al., 2025c;b; Han et al., 2025), or trajectories (Zhao et al., 2024; Fang et al., 2025b). However, none of these representations simultaneously offer strong transferability, efficient retrieval, and direct executability. Inspired by Claude Skills (Anthropic), we argue that **skills** provide a more suitable abstraction: they encapsulate reusable competencies that directly support task execution. Nonetheless, prior skill-based designs often rely on long-context, progressively disclosed specifications, which place heavy demands on reasoning and environment instrumentation, limiting robustness and practical reuse, as illustrated in Figure 1.

In this work, we introduce `AutoSkills`, a fully automated framework for constructing a **plug-and-play skill knowledge base** from agent experience. Our core insight is that transferable experience should be organized **hierarchically**, rather than as monolithic behaviors. `AutoSkills` therefore represents experience at three complementary levels: *(i) planning skills*, which capture high-level task organization; *(ii) functional skills*, which implement reusable, tool-based subroutines; and *(iii) atomic skills*, which encode execution-oriented usage patterns and constraints. This multi-level design yields skills that are concise, composable, and robust to distributional shifts. `AutoSkills` builds such a skill library through a fully automated pipeline. A strong backbone agent first performs rollouts on training tasks and distills multi-level skills from successful trajectories. The extracted skills are then **iteratively refined** through consolidation and validation, improving library quality over time. Finally, `AutoSkills` performs **experience-guided exploration** to proactively expand the skill space by targeting under-utilized tools and failure-prone behaviors, enabling generalization beyond the initial training distribution.

To build a reliable, plug-and-play skill library, we instantiate `AutoSkills` with a strong agent backbone, GLM-4.6 (Team et al., 2025a), and pre-build a skill library on challenging, user-interactive, long-horizon benchmarks, including: AppWorld (Trivedi et al., 2024), BFCL-v3 (Patil et al., 2025), and $\tau^2$-Bench (Barres et al., 2025). Our experiments show that this plug-and-play library can be directly plugged into base agents (e.g., Qwen3-32B (Yang et al., 2025)), yielding around a 10% performance improvement while also improving execution efficiency. We further demonstrate the advantages of our multi-level skill design for experience representation, and show that both iterative refinement and skill expansion provide additional gains. In a nutshell, we conclude our contributions as:

- We propose a hierarchical skill representation that transforms raw trajectories into reusable planning, functional, and atomic skills.
- We present `AutoSkills`, a fully automated and extensible framework for pre-building plug-and-play skill libraries for LLM agents, featuring iterative refinement and skill expansion.
- We release the resulting plug-and-play skill library and provide strong empirical evidence across multiple agent benchmarks that it can directly enhance the capabilities of weaker agents.

## 2. Preliminaries

**Agent Definition** We consider a general interactive setting where an agent solves tasks by acting in an environment. An environment is defined as $\mathcal{E} = (\mathcal{S}, \mathcal{A}, \mathcal{P})$, where $\mathcal{S}$ is the set of observable states, $\mathcal{A}$ the set of executable actions, and $\mathcal{P}(s' \mid s, a)$ the transition dynamics. At time step $t$, the agent receives an observation $o_t \in \mathcal{O}$ and produces an action $a_t \in \mathcal{A}$. Following the ReAct style formulation, the agent therefore selects an action $\hat{a}_t \in \hat{\mathcal{A}}$ conditioned on its context $c_t = (o_1, \hat{a}_1, \ldots, o_{t-1}, \hat{a}_{t-1}, o_t)$:

$$\hat{a}_t \sim \pi(\cdot \mid c_t), \qquad \hat{a}_t \in \hat{\mathcal{A}}. \tag{1}$$

Executing $\hat{a}_t \in \mathcal{A}$ yields a new observation via the environment. The final trajectory is $\tau = (o_1, \hat{a}_1, \ldots, o_T, \hat{a}_T)$.

**LLM Agent and Skill-Conditioned Execution.** Let $\mathcal{Q}$ be the tasks set. We write $q \in \mathcal{Q}$ for sampling a task, and let $R(\tau, q) \in \{0, 1\}$ be a task-dependent success indicator. We model the LLM agent as a policy $\pi$ that induces a trajectory distribution. Without external skills, the agent generates trajectories by direct reasoning:

$$\tau \sim \pi(\cdot \mid q), \qquad q \in \mathcal{Q}. \tag{2}$$

To reduce redundant exploration and improve task completion, we equip the agent with a *skills library* $\mathcal{D} = \{s_1, \ldots, s_{|\mathcal{D}|}\}$ and a *skill retriever* that recalls a set of relevant skills for the current task. Concretely, given $q \in \mathcal{Q}$, a retrieval function (typically implemented via semantic-similarity retrieval) $\rho : \mathcal{Q} \to 2^{\mathcal{D}}$. returns a skill subset $\mathcal{S}_q = \rho(q), \mathcal{S}_q \subseteq \mathcal{D}$. The LLM agent then generates a trajectory by conditioning on the retrieved skill set:

$$\tau' \sim \pi(\cdot \mid \mathcal{S}_q, q), \qquad q \in \mathcal{Q}. \tag{3}$$

Our objective is to design the skills library $\mathcal{D}$ and the usage within $\pi$ such that the expected success rate is improved:

$$\mathbb{E}_{q \in \mathcal{Q}, \, \tau' \sim \pi(\cdot | \mathcal{S}_q, q)} R(\tau', q) \; > \; \mathbb{E}_{q \in \mathcal{Q}, \, \tau \sim \pi(\cdot | q)} R(\tau, q). \quad (4)$$

## 3. **AutoSkills** Design and Implementation

### 3.1. Multi-Level Skills Design

In tool-centric agent scenarios, we structure the skills required by the model into three levels (see Figure 2):

$$\mathcal{D} = S_{\text{plan}} \oplus S_{\text{func}} \oplus S_{\text{atomic}}, \quad (5)$$

corresponding to planning skills, functional skills, and atomic skills, respectively. In a given environment $\mathcal{E}$, let $\mathcal{T}$ denote the set of tool actions. *i) Atomic skill* $s_{\text{atomic}}$ is aligned with a single tool $t \in \mathcal{T}$ and is modeled as an extended semantic specification of $t$, e.g., as enriched descriptions, constraints, or usage patterns that refine the effective behavior of $t$. *ii) Functional skill* $s_{\text{func}}$ abstracts a subtask and can be regarded as a macro-operation that accomplishes a sub-query. We assume each task $q$ admits a decomposition into $n$ subtasks, $\{q_{\text{subtask},1}, q_{\text{subtask},2}, \ldots, q_{\text{subtask},n}\}$ and each $s_{\text{func}}$ corresponds to skills to accomplish $q_{\text{subtask},i}$. Specifically, $s_{\text{func}}$ is grounded in a set of tool actions, which can be instantiated as a composition of tools $\mathcal{T}_{\text{func}} \subseteq \mathcal{T}$. *iii) planning skill* $s_{\text{plan}}$ aligns with the organizational structure of the subtasks (e.g., ordering, dependencies, and branching), specifying how functional skills should be composed to solve $q$. Next, we describe the extraction methods for the three skill levels.

### 3.2. Rollout and Skills Extraction

Given a task $q$, we first perform $m$-sized rollouts, reusing the agent's inference procedure to collect trajectories. We then extract the multi-level skills from these trajectories, with skill extractor $f$. Details of the inference procedure are provided in Section 4.

**Planning Skills Extraction.** Given a successful trajectory, we extract the planning skill $s_{\text{plan}}$ by compressing the trajectory into an ordered set of high-level steps that capture subtask structure. During this compression, we explicitly filter out non-essential transitions such as exploration, backtracking, and trial-and-error behaviors that are incidental to the final solution but detrimental to skill reuse. Moreover, for excessively long or verbose environment feedback, we apply summarization to obtain compact state descriptions, which improves the stability and fidelity of the extracted high-level skills.

**Functional Skills Extraction.** We leverage the previously extracted planning skill $s_{\text{plan}}$ to guide the extraction of functional skills. Concretely, given a plan and its corresponding

trajectory, we iteratively prompt the model to extract the functional skill $s_{\text{func}}$ that aligns with the objective of each subtask $q_{\text{subtask},i}$. Formally, each $s_{\text{func}}$ is represented with three key fields: `name` (the skill name), `document` (a description of inputs, outputs and usage notes), and `content` (the tool invocation pattern for completing subtask $q_{\text{subtask},i}$).

**Atomic Skills Extraction.** Atomic skills are single tool specifications that extend the original tool schema with reusable, execution-oriented usage patterns. They serve as a low-level complement when higher-level functional skills $s_{\text{func}}$ are missing or incomplete. We prompt the model to distill $s_{\text{atomic}}$ from trajectories the invocation patterns, typical parameter configurations, and practical notes, especially constraints and common failure modes observed in real usage. The representation of $s_{\text{atomic}}$ is unified with $s_{\text{func}}$.

### 3.3. Iterative Skills Refinement

With only a limited amount of seed training data, a key question is whether we can maximize the utility of the available supervision to extract additional skills and continuously improve existing ones. Inspired by prior works (Cai et al., 2025b;a; Yuksekgonul et al., 2024), we adopt a text-based iterative optimization paradigm for the skill library. Concretely, at $k$-th iteration, we start from the current skill library $\mathcal{D}^{(k)}$, repeatedly rollouts from the training set, then extract multi-level skills. We subsequently apply a refinement operator $\phi$, including: *Skills Merge* and *Skills Filter*. Finally, we update the skill library $\mathcal{D}^{(k)}$ with the refined skills to obtain skill library $\mathcal{D}^{(k+1)}$, including three update operations: `add`, `modify` or `keep`.

**Iterative Skills Library Construction.** We construct the skill library in an iterative manner. Let $\mathcal{D}^{(0)} = \emptyset$ be an initial empty library. In iteration $k = 0, 1, \ldots$, we roll out the agent augmented with the current library $\mathcal{D}^{(k)}$ on tasks sampled from the training set $\mathcal{Q}_{\text{train}}$ to obtain a set of trajectories

$$\tau^{(k)} \sim \pi(\cdot \mid \rho_{\mathcal{D}^{(k)}}(q), q), \quad q \in \mathcal{Q}_{\text{train}} \quad (6)$$

and denote $\mathcal{K}^{(k)} = \{\tau_1^{(k)}, \ldots, \tau_{N_k}^{(k)}\}$. A skill extractor $f$ produces a variable-size set of candidate skills from each trajectory, $\mathcal{S}_i^{(k)} = f(\tau_i^{(k)})$ and we aggregate all the skills extracted from the batch via $\mathcal{S}^{(k)} = \bigcup_{i=1}^{N_k} \mathcal{S}_i^{(k)}$. Additionally, we define a refinement operator $\phi$ to merge and filter the skills. The library is then updated as

$$\mathcal{D}^{(k+1)} \triangleq \mathcal{D}^{(k)} \cup \phi\Big(\mathcal{S}^{(k)}\Big) = \mathcal{D}^{(k)} \cup \phi\left(\bigcup_{i=1}^{N_k} \mathcal{S}_i^{(k)}\right) \quad (7)$$

Let $\mathcal{Q}_{\text{test}}$ denote a test distribution. We aim to iteratively improve the library such that the performance of the induced

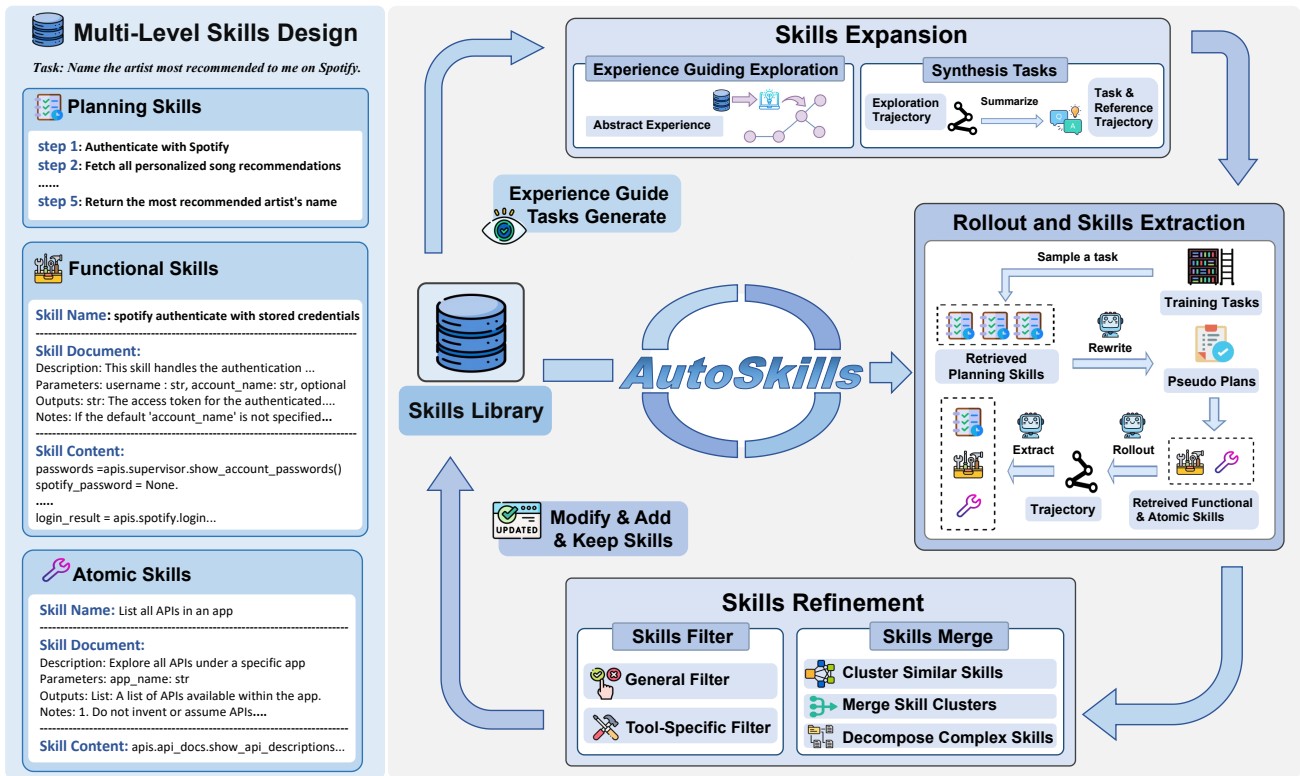

*Figure 2.* AutoSkills provides an automated, iterative pipeline for constructing a skills library, integrating skills extraction. skills expansion and skills refinement. The skills library is organized into three levels: planning skills, functional skills, and atomic skills.

skill-conditioned agent is maximized on $\mathcal{Q}_{\text{test}}$:

$$\max_{k}\ \mathbb{E}_{q\sim\mathcal{Q}_{\text{test}}}\Big[\mathbb{E}_{\tau\sim\pi(\cdot|\rho_{\mathcal{D}^{(k)}}(q),q)}\big[R(\tau,q)\big]\Big], \quad (8)$$

and we stop the iteration when this test performance no longer improves.

**Skills Merge.** After extracting skills from each trajectory, we often obtain many functionally redundant skills that, despite surface differences, correspond to the same underlying skill pattern. How to update a single skill when multiple heterogeneous update directions are available? We merge skills from an optimization-based perspective. For a specific skill $s$ with current embedding, we first retrieve and cluster a set of semantically similar skills using cosine similarity. The resulting cluster can be interpreted as providing multiple complementary update directions for the same underlying skill, a multi-dimensional refinement of $s$. Let $\mathcal{Z}(s) = \{1, \ldots, z\}$ index the semantically similar skills associated with skill $s$. Each neighbor $i$ induces a candidate update direction $\delta_i$, yielding a candidate updated state

$$s'_i\ =\ s + \delta_i, \qquad i \in \mathcal{Z}(s). \quad (9)$$

We then aggregate these candidate directions into the final direction. The simplest form is to merge the directions: $\delta_{\text{agg}}\ =\ \sum_{i\in\mathcal{Z}(s)}\delta_i$. The final update is applied as

$$s^{+}\ =\ s + \delta_{\text{agg}}, \quad (10)$$

In words, we treat the semantically similar skills as multiple update views of the same skill, and we use the combined direction as the final update direction. Finally, we merge semantically similar skills into a single skill. If the merged skill becomes overly complex, we further decompose it into more modular, reusable skills.

**Skills Filter.** We enforce skill quality via a strict two-stage filtering procedure. *1) General Filter* removes skills that are unlikely to be portable or compositional, including those that depend on extraneous Python packages, expose overly idiosyncratic function-style definitions, or overly-encapsulated skills. *2) Tool-specific Filter* mitigates tool-use hallucinations by validating each skill against the environment-provided tool schema, rejecting skills that reference non-existent tools, invalid parameters, or schema-incompatible argument structures.

**Skills Library Update.** After *Skill Merge* and *Skill Filter*, we perform concrete updates to the skill library $\mathcal{D}^k$ for the $k$-th iteration, including three types: add new skills, modify existing skills, and keep skills unchanged. Additionally, the pipeline can be executed iteratively over multiple rounds to continuously refine the skill library.

### 3.4. Exploratory Skills Expansion

While skills distilled from a seed training set $\mathcal{Q}_{\text{train}}$ can already improve an agent's performance, relying solely on scarce demonstrations is insufficient in complex environments with large tool spaces (e.g., (Trivedi et al., 2024) exposes hundreds of APIs). Inspired by (Zhai et al., 2025), we adopt an *Experience Guiding Exploration* scheme to broaden coverage beyond what is observed in the seed data, encouraging the agent to interact with the environment and exercise a wider range of tools. We guide exploration using experience collected from rollouts on the seed set (e.g., tools the agent already uses reliably, tools with high failure rates, and tools that are never invoked), thereby prioritizing under-explored or failure-prone tools to improve sample efficiency. After collecting exploratory trajectories, we synthesize new tasks $\mathcal{Q}_{\text{syn}}$ from these interactions, and then rerun our skill acquisition and refinement pipeline on the resulting data to iteratively expand the skill library. Compared to the random exploration strategy (Zhai et al., 2025), our approach discovers a more diverse set of skills.

## 4. AutoSkills Usage

**Planning Skills Retrieve and Pseudo-Plan Rewrite.** For a novel and complex agent task $q$, directly recalling past experience by task similarity can lead to a mismatch between retrieved experiences and the actual execution trajectory, especially in environments strongly influenced by user profiles or other factors. To improve retrieval relevance, inspired by (Gao et al., 2022), we first retrieve high-level plans associated with similar tasks $\mathcal{P}(q) = \rho(q)$, where $\rho$ is a similarity retrieval function and $\mathcal{P}(q)$ is the retrieved planning skills. Then we prompt the model to self-rewrite a task-specific pseudo-plan conditioned on the current task $\tilde{p}(q) = \text{LLM}_{\text{rewrite}}(q, \mathcal{P}(q))$. This rewritten pseudo-plan serves as an intermediate retrieval query to better align subsequent skill retrieval with the current execution setting. To mitigate hallucination risks and prevent speculative content from affecting agent behavior, the pseudo-plan is not injected into the final system prompt.

**Functional and Atomic Skills Retrieve.** Given the rewritten pseudo-plan $\tilde{p}(q) = \{\text{step}_1, \text{step}_2, \ldots, \text{step}_p\}$, we treat each step as a retrieval query to retrieve functional and atomic skills. For $\text{step}_i$, we first retrieve relevant skills $\mathcal{S}_i = \rho(\text{step}_i)$ and then remove duplicates across steps,

$\mathcal{S}' = \text{dedup}\left(\bigcup_{i=1}^p \mathcal{S}_i\right)$. To keep the context concise and task-relevant, we further ask the LLM to self-filter the retrieved candidates and retain only applicable skills $\mathcal{S}_q = \text{LLM\_select}(q, \tilde{p}(q), \mathcal{S}')$, where $\mathcal{S}_q$ is the final skill set used for solving the query $q$.

## 5. Experiment

### 5.1. Experimental Settings

**Benchmarks and Metrics.** We conduct the evaluation on complex, long-horizon, user-interactive agent benchmarks, including BFCL-v3 (Patil et al., 2025), AppWorld (Trivedi et al., 2024), and $\tau^2$-bench (Barres et al., 2025). For BFCL-v3, we use the base multi-turn category and randomly split it into 50 training instances and 150 test instances. AppWorld provides 90 training instances and the Test Normal category as test set. $\tau^2$-bench defines training and test splits for each sub-domain. Additional details are provided in the Appendix A.1. For AppWorld and BFCL-v3, we report Avg@4 and Pass@4, the average success rate over four independent runs and the probability of succeeding at least once across four runs, respectively. Following the (Barres et al., 2025) evaluation setup, we report Pass^1, the pass rate over running four times.

**Models and Baselines.** To assess the effectiveness of AutoSkills, we evaluate three Agentic base models that vary in model size and reasoning style (thinking and non-thinking), including Qwen3-32B (Yang et al., 2025), Kimi-K2-Instruct-0905 (Team et al., 2025b), and GLM-4.6 (Team et al., 2025a). Among them, GLM-4.6 has been reported to exhibit strong native agentic capabilities in agent mid-training, serving as a competitive backbone for our study.

We compare against four representative baselines: (1) No-memory, which performs inference without retrieving any prior experience; (2) A-Mem (Xu et al., 2025), a system that dynamically manages structured episodic memories; (3) AWM (Wang et al., 2025c), which reuses modular workflows distilled from historical trajectories; and (4) ExpeL (Zhao et al., 2024), which retrieves relevant past trajectories as few-shot demonstrations and incorporates distilled insights to improve LLM performance. For a fair comparison, all methods retrieve experience only based on the user's initial query and insert the retrieved content into the system prompt following a unified protocol. Full baseline details are provided in the Appendix A.2.

**Implementation Details.** To construct AutoSkills, we use GLM-4.6 (Team et al., 2025a) independently roll-outs four times per training task, followed by skill extraction, skill refinement, and skill expansion. The maximum number of refinement iterations is set to 3. For efficiency, we limit environment exploration to one rollout per training task;

| Model | Methods | BFCL-V3 | | AppWorld | | $\tau^2$-Bench | | |
|---|---|---|---|---|---|---|---|---|
| | | Avg@4 | Pass@4 | Avg@4 | Pass@4 | Retail | Airline | Telecom |
| **Qwen3-32B** | No Memory* | 53.67 | 73.33 | 27.68 | 47.62 | 53.75 | 38.75 | 36.25 |
| | A-Mem* | 53.67 | 73.00 | 26.79 | 50.59 | 53.12 | 38.75 | 38.12 |
| | AWM* | 55.67 | 76.00 | 30.80 | 55.95 | 55.00 | 40.00 | 38.12 |
| | AWM‡ | 56.67 | 76.33 | 34.45 | 56.25 | 57.50 | 41.25 | 40.62 |
| | ExpeL* | 57.33 | 77.67 | 32.87 | 58.93 | 56.25 | 42.50 | 39.38 |
| | ExpeL‡ | 59.33 | 78.83 | 32.94 | 58.78 | 58.12 | 43.75 | 41.25 |
| | AutoSkills‡ | **63.67** | **82.00** | **35.12** | **58.93** | **66.87** | **47.50** | **43.75** |
| **Kimi-K2-Instruct-0905** | No Memory* | 65.17 | 78.00 | 46.88 | 70.24 | 75.62 | 51.25 | 78.12 |
| | A-Mem* | 65.17 | 76.67 | 46.58 | 72.62 | 76.25 | 52.50 | 76.87 |
| | AWM* | 65.33 | 79.00 | 49.70 | 76.19 | 76.25 | 53.75 | 77.50 |
| | AWM‡ | 64.67 | 79.17 | 50.60 | 76.49 | 76.25 | 53.75 | 77.50 |
| | ExpeL* | 66.33 | 79.33 | 52.53 | 78.57 | 77.50 | 55.50 | 78.75 |
| | ExpeL‡ | 66.00 | 79.67 | 52.98 | 78.87 | 77.50 | 56.25 | 79.37 |
| | AutoSkills‡ | **66.83** | **81.33** | **56.40** | **81.55** | **78.12** | **58.75** | **82.50** |
| **GLM-4.6** | No Memory* | 76.67 | 83.33 | 60.27 | 83.33 | 76.25 | 70.00 | 70.63 |
| | A-Mem* | 76.50 | 83.00 | 60.57 | 83.93 | 76.88 | 70.00 | 68.75 |
| | AWM* | 77.17 | 84.00 | 62.20 | 84.52 | 77.50 | 71.25 | 70.63 |
| | ExpeL* | 78.83 | 85.33 | 64.14 | 85.12 | 77.50 | 72.50 | 71.25 |
| | AutoSkills* | **79.50** | **86.00** | **64.88** | **88.69** | **82.50** | **76.25** | **71.88** |

*Table 1.* Main Results of AutoSkills on three benchmarks. Methods with ∗ indicate that the experience extraction model is aligned with the inference model. Methods with ‡ indicate that GLM-4.6 is used for experience extraction, while inference still relies on the original model.

the sampling temperature is 1.0 during exploration and 0.9 otherwise. We use Qwen3-Embedding-8B (Zhang et al., 2025d) for both skill deduplication and skill retrieval, with a minimum cosine similarity threshold of 0.45 for retrieval. During solving new tasks, we use the same model for both Pseudo-Plan rewriting and action execution. For the other baselines, we keep the experience extraction model consistent with the execution model to support self-extraction, and we follow the original experimental protocols of the corresponding methods. For a fair comparison, we only evaluate the setting with a pre-built experience repository. Additional implementation details are provided in the Appendix A.3.

### 5.2. Main Results

**AutoSkills Boost Agentic Performance of Base LLMs.** As shown in Table 1, AutoSkills improves the base model's performance. In particular, Qwen3-32B gains roughly around 10 points across multiple benchmarks. For K2 (Kimi-K2-Instruct-0905), we observe a clear improvement on AppWorld, whereas the gains are modest on the other two tool call intensive benchmarks. We conjecture that this is because K2 relies more heavily on the original tool schema and does not effectively leverage the additional contextual information.

**Multi-Level Skills Design Outperform Other Forms of Experience Representation.** When the experience extraction model is aligned with the execution model, AutoSkills consistently outperforms all baseline methods, as indicated by the methods with ∗ in Table 1. Among them, ExpeL retrieves past trajectories and uses them as few-shot demonstrations, which provides a more direct performance gain than the other baselines.

**Suboptimal Experience Representations Hinder Transfer Performance.** We further evaluate the GLM-4.6 extracted experience from AWM and ExpeL on the weaker models, see the results of methods with ‡ in Table 1. However, the performance still lagged behind that of AutoSkills. This indicates that **distilling experience from a strong model is effective, but the form of experience representation is even more critical**. Consequently, suboptimal experience representation can hinder effective experience transfer. These results further demonstrate the advantage of AutoSkills in transferring experience across base models.

**AutoSkills can Expand Base Model's Capability Boundary.** We observe that experience-based learning leads to substantial Pass@4 improvements for the weaker

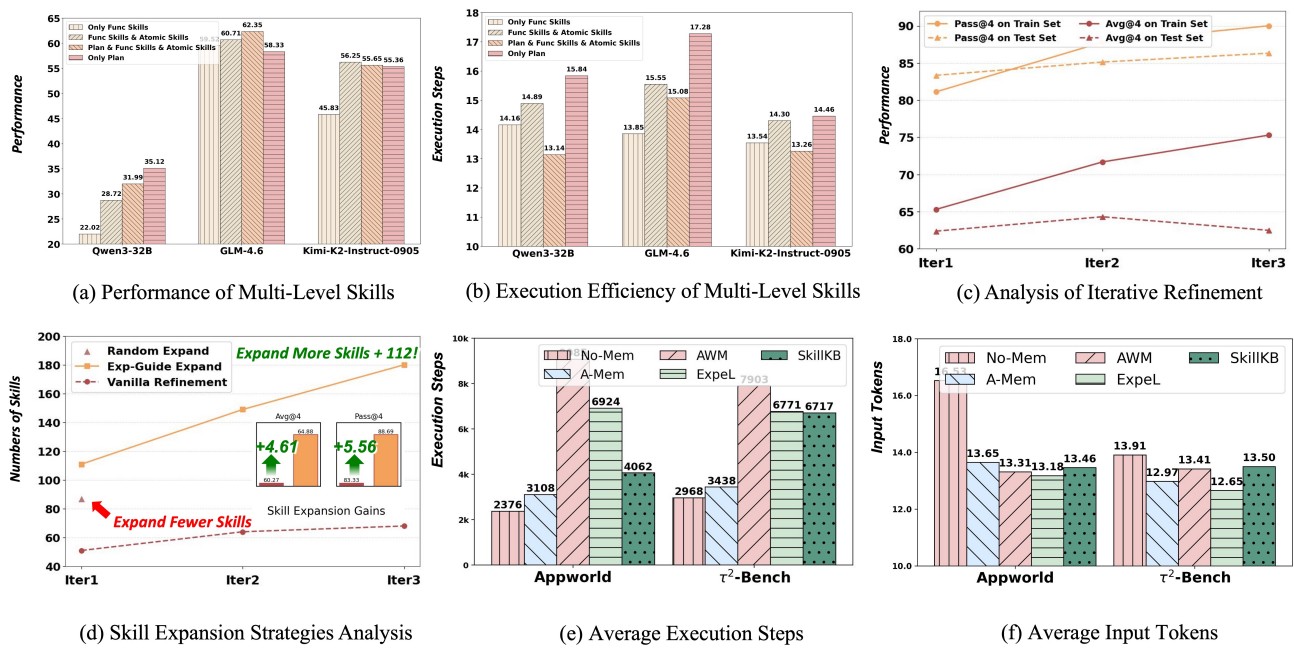

*Figure 3.* **Comprehensive Analysis of AutoSkills. (a) Performance of Multi-skills:** Models exhibit varying performance under different skill composition. **(b) Execution efficiency of Multi-skills:** Jointly composing all skills yields the best execution efficiency. **(c) Iterative optimization:** Iterative skill refinement further improves performance. **(d) Skill expansion strategies:** Experience-guided expansion achieves the best on scalability and performance gains. **(e) Analysis of Input tokens:** Properly balancing input tokens is crucial for controlling inference cost. **(f) Analysis of Execution steps:** Experience-based learning reduces the number of execution steps.

models, K2 and Qwen3-32B. This suggests that, in practice, the most direct way to extend the capability boundary of a base model is to distill knowledge from a stronger model (Yue et al., 2025). In contrast, for the stronger agent model GLM-4.6, neither the baseline nor AutoSkills yields a significant gain in Pass@4. This indicates that stronger models already possess robust capabilities in exploration, planning, and tool use, leaving limited headroom for further capability expansion via experience-based augmentation. Nevertheless, the modest improvements still support the effectiveness of AutoSkills.

### 5.3. Analysis

**Which skill is more effective?**   We analyze the behaviors of our multi-level skill across models on AppWorld, with results shown in Fig. 3 (a) and Fig. 3 (b). *i) Planning skills* consistently reduce the number of execution steps across all models, with particularly pronounced gains for weaker models such as Qwen3-32B and K2, especially when combined with Functional Skills. We attribute this to their limited exploration capability in complex environments. Notably, for Qwen3-32B, adding Functional and Atomic Skills can even hurt performance, as the model tends to over-imitate retrieved skills rather than adapt them to novel tasks. For

stronger models, pseudo-planning may fail to faithfully capture underlying environment dynamics in complex scenarios, and can therefore become counterproductive. *ii) Functional skills* contribute the most to overall performance improvements: equipping K2 and GLM-4.6 with Functional and Atomic Skills alone already yields observable gains, highlighting the advantage of skills as an effective representation of experience. *iii) Atomic skills* provide crucial clarifications for key APIs. When they are absent, performance drops substantially, further validating the need to supplement tool schemas and to cover tools missing from Functional Skills. Finally, we find that GLM-4.6 benefits most from using all skill types; K2 performs best with Functional + Atomic Skills; and Qwen3-32B achieves its best performance when only Planning Skills are enabled. This further demonstrates that multi-level skills can comprehensively cover the capabilities required for diverse models to execute agent tasks.

**Iterative Refinement Strategies Further Enhances AutoSkills Performance.**   We evaluate the effectiveness of multi-round iterative refinement of the skill library on AutoSkills in AppWorld (see Figure 3 (c)). Overall, multiple iterations further improve performance on both the training and test sets. Leveraging existing training data, the process continually improves various aspects of the skills,

including document and content. Meanwhile, it can slightly expand the size of the skill library (see Figure 3 (d)). However, when training data are limited, text-only optimization may lead to overfitting. Therefore, selecting an appropriate number of update rounds is crucial for obtaining a higher-quality skill library.

**Skill Expansion Strategies Improve Generalization.** We compare two skill expansion strategies: *random exploration* and *experience-guided* expansion. The results are as shown in Figure 3 (d). In terms of skill growth, the experience-guided strategy yields substantially more novel skills, as random exploration treats past executions in isolation and repeatedly rediscovers already identified skills. Empirically, the experience guided strategy yields performance improvement through skill expansion. Overall, our results indicate that in complex environments, particularly under scarce training data, skill expansion is a crucial component of experience learning.

**AutoSkills Enhances Agent Execution Efficiency.** Learning from experience not only improves the performance of the base model, but also enhances the execution efficiency of the agent. Our experiments further corroborate this effect (see Figure 3 (e) and Figure 3 (f)). Although we do not achieve the minimum number of execution steps or the fewest input tokens, we obtain the best overall performance (see Table 1). These results further highlight the advantages of our multi-level skill design and skills library construction.

**Case Study.** We provide qualitative cases illustrating how agents leverage AutoSkills and how retrieved skills shape their behavior when solving unseen tasks. Detailed cases are included in the Appendix B.

## 6. Related Work

**Encoding For Agent Experience.** With the advent of the experience era (Sutton, 2025) agents can achieve self-evolving (Gao et al., 2025; Fang et al., 2025a) by encoding past experience and reusing it to guide future behavior. Existing approaches to text token-level experience encoding (Zhang et al., 2025b; Hu et al., 2025) can be broadly grouped into three categories: *i) Case-based Experience*: Agents directly store successful task-execution trajectories and retrieve them later as few-shot examples to new problem solving (Zhao et al., 2024; Zheng et al., 2024; Zhou et al., 2025). *ii) Strategy-based Experience*: By summarizing and contrasting successful versus failed trajectories, agents distill higher-level insights or workflows (Cao et al., 2025; Ouyang et al., 2025; Cai et al., 2025a; Wang et al., 2025c; Tang et al., 2025; Zhang et al., 2025a). *iii) Skill-based Experience*: Trajectories are segmented and distilled

into modular, reusable skills, such as textual skills or programmatic skills (Wang et al., 2025b;a; 2024; Fang et al., 2025b; Han et al., 2025). However, it remains unclear which unified experience representation is both easily pluggable and consistently effective, especially in diverse and complex agentic tool-use scenarios (Trivedi et al., 2024; Yao et al., 2024; Patil et al., 2025; Barres et al., 2025; He et al., 2025; Li et al., 2025). In this work, we adopt a hybrid representation, high-level planning coupled with textual skills, which yields substantial improvements for the base model.

**Agent Experience Knowledge Base Construction.** The construction pipeline of an experience knowledge base typically consists of two steps: static construction and dynamic updating. *i) Static construction* repeatedly attempts tasks on a training set, extracts experience, and iteratively refines it until performance plateaus (Zhang et al., 2025c; Cai et al., 2025b;a). *ii) Dynamic updating* updates the ExperienceKB immediately after executing new tasks, enabling experience reuse in subsequent tasks (Cao et al., 2025).

While dynamic updating is central to continual learning from experience, pre-building a strong static ExperienceKB remains necessary in practice. However, under the task-scarcity challenge in complex agent settings (Patil et al., 2025; Barres et al., 2025; He et al., 2025; Li et al., 2025), we further extend skills by combining task synthesis (Zhai et al., 2025; Mai et al., 2025; Shi et al., 2025; Ramrakhya et al., 2025) to construct more challenging tasks. To our knowledge, this is the first work to provide a directly reusable skill knowledge base together with an automated pipeline for skill construction.

## 7. Conclusion

We introduced AutoSkills, an automated framework for building a plug-and-play skill library for LLM-based agents. To enable more efficient experience transfer, we design a multi-level skills, including planning skills, functional skills, and atomic skills from the perspective of tool granularity. AutoSkills iteratively refines and expands the library through three core components: *i) skills extraction*, which rolls out an agent with the current library and extracts multi-level skills; *ii) skills refinement*, which iteratively improves skills using execution feedback, while maintaining quality via skill merging and strict filtering; and *iii) exploratory skills expansion*, which proactively broadens coverage beyond the seed training set. Our experiments demonstrate that AutoSkills transfers effectively to other models and provides advantages in experience representation. Finally, we will release the optimized skill library constructed by AutoSkills to facilitate further community exploration.

## Impact Statements

This work advances generalizable agent learning by transforming isolated trial-and-error experience into a reusable, structured skill knowledge base that can be shared across agents and environments. By enabling weaker agents to benefit from skills distilled by stronger ones, the proposed framework reduces redundant exploration, improves sample efficiency, and lowers the computational and environmental costs of training LLM agents. The plug-and-play design promotes modularity and reproducibility, supporting broader adoption in long-horizon, user-interactive applications. Potential risks include over-reliance on pre-built skills and the propagation of biases present in source agents; however, the automated refinement and expansion mechanisms provide a pathway to mitigate stagnation and encourage continual adaptation.

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

# A. Detailed Experiments Settings

## A.1. Benchmark Details

**BFCL-v3**  Berkeley Function Calling Leaderboard V3 (BFCL-v3) (Patil et al., 2025) is a benchmark for evaluating function calling and tool use in large language models. It emphasizes multi-turn interaction and multi-step reasoning. The benchmark contains over 1,800 test instances and supports multiple programming languages, including Python, Java, and JavaScript. Models are required to generate valid API calls and handle non-trivial interaction patterns. Evaluation considers both structural validity and functional correctness. We first check whether the generated code is syntactically valid using Abstract Syntax Tree analysis, and then execute it to verify that the outputs match the expected results. A task is considered successful only when the agent produces all required function calls with correct syntax and returns the correct computational outcomes. In this work, we report Avg@4, which measures the average task success rate across four independent trials, and Pass@4, which measures the probability that at least one of the four trials succeeds.

**Appworld**  AppWorld (Trivedi et al., 2024) is a benchmark suite for evaluating function calling agents and interactive coding systems in realistic application environments. It simulates an ecosystem of nine widely used applications, such as email services, music streaming platforms, and payment systems, and provides 457 API endpoints together with activity data from around 100 virtual users. Tasks in AppWorld are typically long-horizon and require executing extended sequences of interdependent actions. Many tasks involve discovering appropriate APIs rather than directly reusing familiar patterns, which places additional demands on exploration and planning. The benchmark also exhibits a noticeable distribution gap between training and test sets, where API usage patterns and task structures in the test set differ from those observed during training. In addition, task execution is tightly coupled with the evolving environment state. Intermediate actions modify the system state and influence future decisions, which increases sensitivity to planning errors and makes robust multi-step reasoning more difficult. Evaluation is based on state-driven unit tests that assess task completion from multiple aspects. AppWorld provides both task-level and scenario-level metrics. In this work, we use Task Goal Completion as the primary measure of performance. Following the standard protocol, we report Avg@4 and Pass@4 across four independent trials.

$\tau^2$**-Bench**  $\tau^2$-Bench (Barres et al., 2025) evaluates tool use in conversational agent settings, with a strong emphasis on user-agent interaction. The benchmark simulates multi-turn dialogues between a user and an agent, aiming to reflect realistic conversational behavior. Agents must track dialogue context across turns, interpret user requests, select and invoke APIs appropriately, and follow domain-specific business rules. The tasks cover domains such as airline customer service and retail customer service. The interactive nature of the benchmark requires agents to respond to user feedback, maintain coherent dialogue flow, and coordinate tool use with the ongoing conversation. Performance is assessed based on task completion accuracy, correctness of tool use, and compliance with policies. In this work, we conduct four independent trials per task and report Pass@1 on each of the three domains.

## A.2. Baseline Details

**A-Mem**  A-Mem (Xu et al., 2025) is an agentic memory framework that equips LLM-based agents with the ability to maintain and utilize long-term knowledge over extended interactions. The method organizes accumulated experiences into a memory-centric structure, enabling agents to selectively retain, retrieve, and revise stored information according to task objectives and observed outcomes. Rather than treating memory as a passive log, A-Mem emphasizes autonomous memory management driven by the agent's goals and interaction context. In our experiments, we reproduce A-Mem based on its publicly available implementation, with minor prompt adaptations to support memory writing and organization during task interactions.

**AWM**  AWM (Agent Workflow Memory) (Wang et al., 2025c) is a memory-augmented agent framework that focuses on discovering reusable workflow patterns from past task executions. The method stores completed task trajectories as episodic experiences and derives higher-level procedural knowledge by analyzing multiple successful examples. Experience retrieval follows a lightweight lexical matching strategy. Textual representations of task queries and stored experiences are mapped to sparse term-based vectors, and relevance is measured using cosine similarity. A small set of highly relevant experiences is selected for downstream analysis, with subsampling applied when multiple candidates exhibit comparable similarity. Workflow induction is performed by prompting a language model to analyze the retrieved successful trajectories and summarize recurring action patterns. Rather than relying on explicit symbolic rules or predefined workflow schemas, AWM captures reusable procedural structures directly from empirical task executions. Retrieved experiences are incorporated

as conversational message objects (e.g., `HumanMessage` and `AIMessage`), enabling the language model to process exemplar interactions naturally within the dialogue context.

**ExpeL** ExpeL (Zhao et al., 2024) is an experience-driven learning framework that improves agent performance by reflecting on past successes and failures. The method stores task execution trajectories and generates experiential knowledge by contrasting successful and unsuccessful outcomes for the same task. In our experiments, we reproduce ExpeL by collecting both successful trajectories (reward $\geq 1.0$) and failed trajectories (reward $< 1.0$). For each successful example, a small number of failed trajectories from the same task type are selected for comparative analysis. A large language model is prompted to analyze the paired trajectories and generate natural-language critiques that highlight key decision differences and improvement suggestions. These critiques are retained as unstructured textual experiences and reused as guidance in subsequent tasks.

**Other Baseline** We also observe that insight-style experience (Cao et al., 2025) can improve agent performance, but it is less effective than ExpeL; therefore, we do not further include it in subsequent experiments.

### A.3. Implementation Details

For hyperparameter settings, we use the default temperature value of $0.5$ for Qwen3-32B and GLM-4.6 models, $0.0$ for Kimi-K2-Instruct-0905. The top-p parameter is set to $0.95$ (the default value) for all models across all experiments.

### B. Case Study For **AutoSkills**

We present case studies across three diverse benchmarks: AppWorld (Trivedi et al., 2024), BFCL (Patil et al., 2025), and $\tau^2$-bench (Barres et al., 2025). These cases show that skill libraries help agents avoid common failures such as incorrect API call sequences, missing prerequisite checks, and the inability to handle conversational topic shifts. By framing domain knowledge as reusable skills, agents can complete complex multi-step tasks that the baseline method fails, reducing trial and error from multiple failed attempts to successful execution on the first attempt.

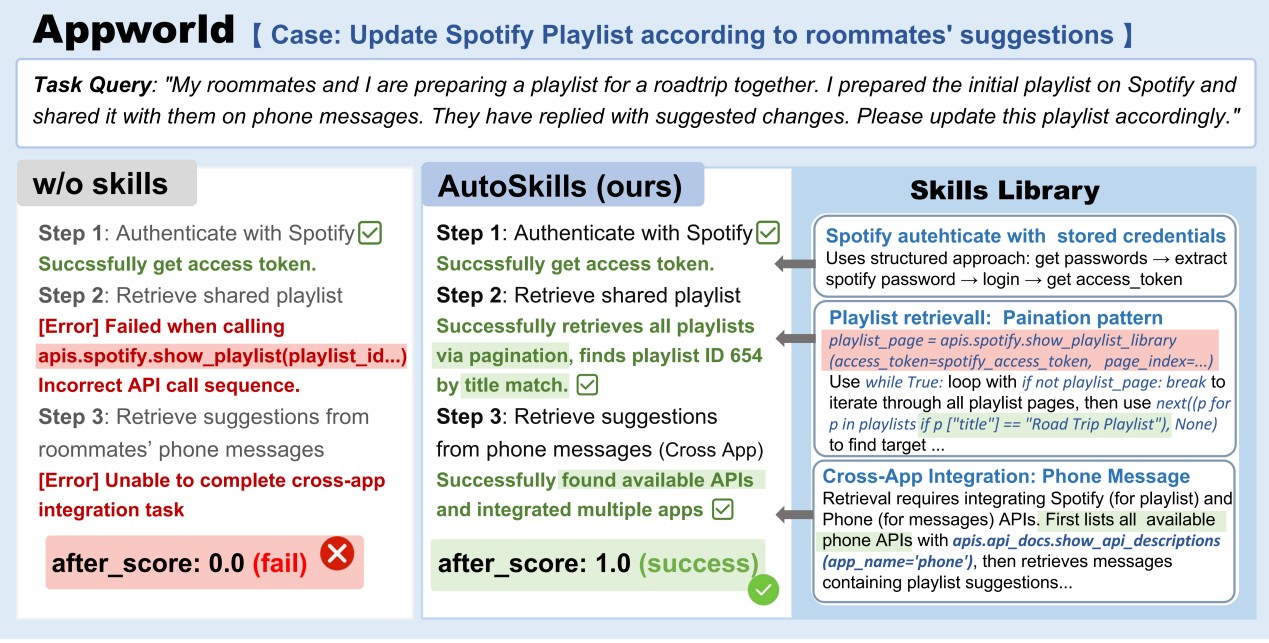

*Figure 4.* **AppWorld benchmark case study: Updating Spotify playlist based on roommates' suggestions.** `AutoSkills` successfully handles API call sequences (pagination pattern for playlist retrieval) and cross-app integration (integrating Spotify and Phone APIs), while the baseline without multi-level skills fails due to incorrect API call sequences and inability to complete cross-app integration tasks.



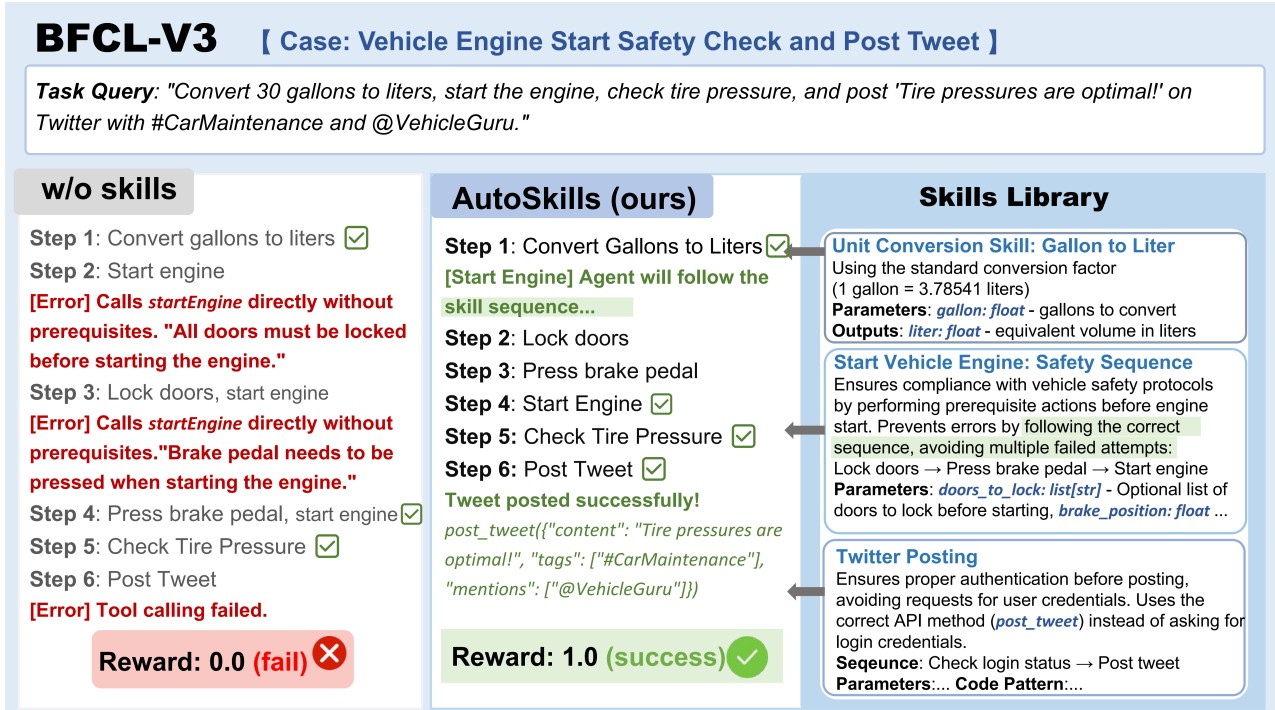

*Figure 5.* **BFCL benchmark case study: Vehicle engine start safety check and Twitter posting.** `AutoSkills` follows prerequisite sequences (lock doors → press brake pedal → start engine) and properly authenticates before posting tweets, while the baseline without multi-level skills fails by calling APIs without prerequisites and encountering tool calling errors.

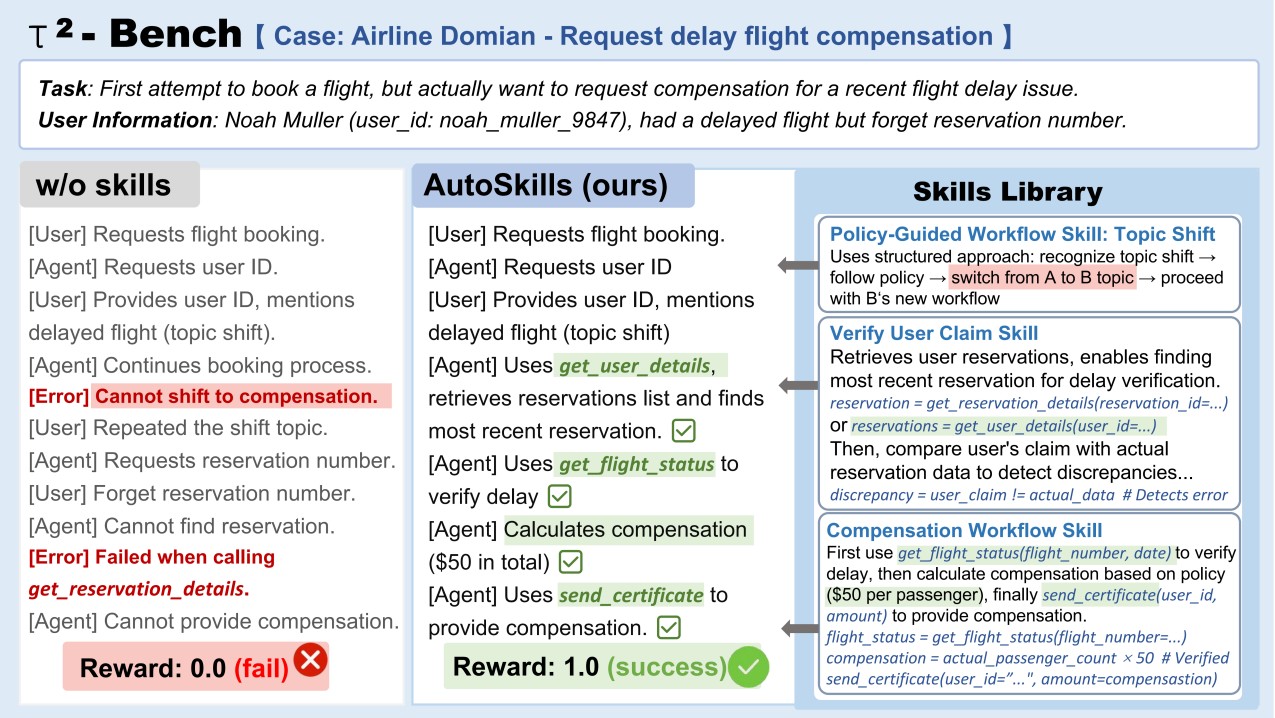

*Figure 6.* $\tau^2$-**bench case study: Requesting delay flight compensation in airline domain.** `AutoSkills` handles topic shifts, retrieves user reservations without reservation numbers, verifies flight delays, and executes the compensation workflow, while the baseline without multi-level skills fails to recognize topic shifts and cannot retrieve reservation details.

## C. Main Prompt Use For `AutoSkills`

In this section, we provide the prompts of `AutoSkills` used for skill extraction, planning, filtering, and merging operations.

### C.1. General Filter Prompt

---

**General Filter Prompt**

You are a coding expert. Given a predefined skill, evaluate whether its quality is good or bad.

**Evaluation guidelines:**
1. **Domain specificity**: Check whether the skill includes domain-specific library names APIs, e.g., {api}.
2. **Over-encapsulation**: Check whether the skill's implementation merely calls a single other skill (i.e., it is just a thin wrapper).
3. **No-Python-libraries**: Check whether additional Python libraries are introduced in the skill.
4. **Reusability**: Check whether the parameters are specific.
5. **No-Functional style**: Check whether a functional style is being used (e.g., the presence of return).

**Bad Example1**
{example}
**Bad Example2**
{example}
**Good Example**
{example}
Only return "good" or "bad". Don't return any other words.

---

*Table 2.* Prompt for filtering skills based on quality criteria.

### C.2. Tool Summary Prompt

---

**Tool Summary Prompt**

You are an AI assistant specialized in analyzing agent trajectories.
Your task is to summarize a single interaction: based on the environment feedback from the current step, extract and summarize the key information in no more than 50 words.

**Inputs Description**
1. The AI assistant's reasoning and action
2. The resulting environment feedback after the action

**Summary Guidelines**
1. Summarize what the environment feedback conveys in light of the AI assistant's intent.
2. Preserve details that are tightly relevant to the intent verbatim when possible; compress other redundant information.
3. Summarize only factual content from the environment feedback—do not invent anything.
4. Write the summary in the tone of the environment feedback.

**Output Format**
<feedback>
Your summary of the environment feedback
</feedback>

---

*Table 3.* Prompt for summarizing environment feedback from agent interactions.

## C.3. Tool Schema Filter Prompt

---

**Tool Schema Filter Prompt**

You are a tool-invocation expert. Based on the tool specifications, verify whether the provided tool invocations are correct.

**Input**
1. **Tool invocation content**: may include one or multiple tool calls.
2. **Tool specifications**: including tool description, parameters, return schema, and other usage notes.

**Judging Guidelines**
1. **Parameter validation**: Check whether the invocation parameters comply with the specifications (e.g., missing required parameters, unsupported/nonexistent parameters, wrong types or formats, invalid values, etc.).
2. **Call dependency**: For multiple tool calls, verify that their order does not violate logical dependencies. If there is no dependency between the calls, ignore this check.
3. **Comment–function alignment**: Ensure the logic described in any comments matches what the tool is designed to do.
4. **Output Format**: Provide your reasoning and conclude with either 'correct' or 'fail', wrapped in <answer></answer>.

---

*Table 4.* Prompt for validating tool invocations against specifications.

## C.4. Plan Extract Prompt

---

**Plan Extract Prompt**

You are a **Planning Expert**.
Your job is to analyze an agent's API interaction history and the user's task, then distill them into a concise, reusable plan. This plan should serve as a reference for handling similar tasks more effectively in the future.

**OBJECTIVES**
1. **Understand Capabilities**
   • Analyze the recorded API calls to identify the actual functional capabilities demonstrated.
2. **Abstract into a Plan**
   • For each feasible task supported by those capabilities, produce a concise, reusable step-by-step plan that can be applied to similar tasks.

**Planning Creation Rules**
**1. Focus**
• Do not simply restate each API function step-by-step using technical jargon. Instead, describe the underlying sub-goal behind each action segment.
**2. Remove Non-Essential Steps**
• Exclude capability exploration, debugging, and failed steps.
**3. Reusability**
• The plan must be precise enough for other models to reuse.
**4. Conciseness**
• Merge steps from the interaction history that share the same objective into a single sub-step in the plan.
• Use a compact writing style for each sub-step, while listing the key APIs involved in that step (one or more).
• Do not omit any critical, potentially required API keys.

**OUTPUT FORMAT**
For each task, output exactly one plan and follow this format strictly:
<plan>
# step 1: A natural, specific, concise sub-task goal; key APIs used (one or more).
# step 2: ...
...
</plan>

**GOOD EXAMPLES**
{examples}

**CHECKLIST BEFORE FINALIZING**
✓ **Reusability** — Ensure no critical steps are missing, and the step order is correct.
✓ **Conciseness** — Confirm there are no redundant or unnecessary steps.
✓ **Agent-centered** — Make sure the plan reads like actionable instructions that other models can reliably follow.

---

*Table 5.* Prompt for extracting reusable plans from agent trajectories.

## C.5. Merge Prompt

> **Merge Prompt**
>
> You are a code expert. Your task is to analyze a list of skills, merge skills that are meaningfully similar, and decompose complex skills into smaller atomic skills while preserving behavior and intent.
>
> **Input Description**
> The user will provide a list of skills.
>
> **Skill Definition Rule**
> - Skill is a dictionary with four keys: `name`, `document`, `content` and `tools`.
>     1. `name`: the skill's name.
>     2. `document`: the skill's functionality, the key parameters, the final output of the skill, and any important notes.
>     3. `content`: the concrete implementation of the skill.
>     4. `tools`: the key tools used in the skill (list).
> - The skill is abstract, modular, and reusable. Specifically, the skill name must be generic under one application (e.g., {good example} instead of {bad example}). The skill must use parameters instead of hard-coded values (e.g., specific email address {email address}). The skill body must be self-contained.
> - Explicitly declare the key parameters and the final output data types using type hints. Example: `Parameters: param: str; Outputs: output: list[dict]:`
> - Include a detailed description of the skill with input and output explanation.
> - The skill should not be similar to the existing skills in the skills library.
> - The skill must involve multiple processing steps. Simply using the result of an API call without additional logic does not qualify as a valid skill.
> - Never call other skills from the skills library or any previously defined skills.
> - Do not import any Python packages.
> - Avoid a functional style; there's no need to use return.
>
> **Good skill:**
> ```json
> {
>     "name": {name},
>     "document": {document},
>     "content": {content},
>     "tools": {tools}
> }
> ```
>
> **Focus**
> 1. Focus on skills with similar names and similar skillality.
> 2. Carefully analyze the concrete implementation differences between similar skills.
>
> **Merge Guidelines**
> 1. **Generality**: Merge skills that have similar names and similar skillality. The merged skill should use a generic name, and its **Notes** and implementation should cover all plausible variants and edge cases.
> 2. **Atomicity**: If skills have a containment relationship (one skill's skillality subsumes or builds on another), follow the skill definitions to preserve atomicity and avoid merging.
> 3. **Merge Constraints**: Any merged skill must comply with the skill definition rules, especially atomicity and reusability, and should avoid being tied to a specific task or scenario.
>
> **Decompose Guidelines**
> 1. **Atomicity**: Only decompose skills whose skillality is overly complex (e.g., they include skillality already covered by other provided skills) into smaller sub-skills.
> 2. **Generality**: The decomposed skills must follow the skill-definition rules and remain reusable—avoid coupling them to any specific task or scenario.
>
> **Output Format**
> Output a list containing the skills (with one or multiple skills) from merging and/or decomposing the skills in the input skill list as follows:
> <skill>
> [
>     "skill 1",
>     ...
> ]
> </skill>
> Note: You don't necessarily need to both merge and decompose. You may choose to only merge them into a single skill.

*Table 6.* Prompt for merging and decomposing skills.

## C.6. Atomic Skill Extract Prompt

---

**Atomic Skill Extract Prompt**

An agent system is provided with a **skills library** and has tried to solve the task multiple times with a successful solution. Review the task-solving attempt and extract generalizable skills.

**1. Inputs Description**
• **User Task**
• **Trajectory**: A record of an agent's interactions successfully with the environment as it attempts to complete a user task.
• **skills library**: A collection of all currently available skills that can be directly reused.
• **Specific-Tool**: Given a specific tool, extract only one reusable skill for the specified tool.

**2. Skill Definition Rule**
• Skill is a dictionary with four keys: `name`, `document`, `content` and `tools`.
    1. `name`: the specific tool's name.
    2. `document`: the tool's functionality, the key parameters, the final output of the skill, and any important notes.
    3. `content`: the tool's usage examples, and examples of combining it with other tools (if applicable).
    4. `tools`: the key tools used in the `content` (list).
• The skill is centered around a specific tool, describing its core functionality, important notes, and common usage examples.
• Explicitly declare the key parameters and the final output data types using type hints. Example: `Parameters: param: str; Outputs: output: dict:`
• Include a detailed description of the skill with input and output explanation.
• The skill should not be similar to the existing skills in the skills library.
• The parameters used in `content` must be reusable instead of hard-coded values (e.g., specific email address "jay@gmail.com")
• The usage examples of `content` may involve one or more tool uses.
• The `document` must clearly and thoroughly document all relevant details of the specific tool use.
• Never call other skills from the skills library or any previously defined skills.
• Do not import any Python packages.
• Avoid a functional style and Python code style; there's no need to use return.

**3. Update Existing Skills**
Your goal is to ensure the system retains actionable skills that help it behave correctly in the future.
You have three options: **[modify, add, keep]**
• **modify**: revise an existing skill to make it more effective (e.g., improving documents). Only change `content` when necessary, and ensure the resulting skill remains broadly general-purpose.
• **add**: introduce a new skill only when the existing skills library is missing the specified tool.
• **keep**: Preserve the skill unchanged when there are no clear issues.
Common actions:
• add a new skill
• update a skill's usage instructions/documentation
• revise a skill's variable/parameter definitions to make it more generalizable
• keep a skill unchanged

**4. Requirements for each skill that is modified or added.**
• **Avoid duplication**: If a skills library is provided, do not add new skills that are similar to existing ones—use **keep** or **modify** instead.
• **Ensure domain specificity**: The skill must contain domain-specific tool.
• **Specific-Tool guided extraction**: Only focus on the specified tool in the trajectory when extracting skills.

**5. Good Skill Example**
{example}

**6. Output Format**
You will finish by returning in this JSON format as follows:
```json
[
  {
    "option": "modify",
    "skill": "the modified skill",
    "modified_from": "spotify get all user playlists" # specify the skill name of existing skills that is modified
  },
  {
    "option": "add",
    "skill": "the added skill",
  },
```

```
    {
      "option": "keep",
      "skill_name": "the kept skill name",
    }, ...
]
```

Note that your updated skills may not need to cover all the options. You can only use one type of updates or choose to remain all skills unchanged.

**7. CHECKLIST BEFORE FINALIZING**
✓ **Reusability** — Ensure no critical steps are missing, each skill is modular, all parameters are abstract rather than specific.
✓ **Optimality** — Ensure each skill meets the required definition standards.
✓ **Agent-centered** — Add helpful notes in each skill to guide other models in using it correctly.
✓ **Specific-Tool focus** — Whether the extracted skill doesn't center around this Tool?

*Table 7.* Prompt for atomic skill extraction based on specific tools.

## C.7. Functional Skill Extract Prompt

**Functional Skill Extract Prompt**

An agent system is provided with a **skills library** and has tried to solve the task multiple times with a successful solution. Review the task-solving attempt and extract generalizable skills.

**1. Inputs Description**
• **User Task**
• **Trajectory**: A record of an agent's interactions successfully with the environment as it attempts to complete a user task.
• **skills library**: A collection of all currently available skills that can be directly reused.
• **Specific-step**: Given a concrete step, extract only one reusable skill for the specified step.

**2. Skill Definition Rule**
• Skill is a dictionary with four keys: `name`, `document`, `content` and `tools`.
    1. `name`: the skill's name.
    2. `document`: the skill's functionality, the key parameters, the final output of the skill and any important notes.
    3. `content`: the concrete implementation of the skill.
    4. `tools`: the key tools used in the skill (list).
• The skill is abstract, modular, and reusable. Specifically, the skill name must be generic under one application (e.g., `spotify get songs by genre` instead of `get pop songs`). The skill must use parameters instead of hard-coded values (e.g., specific email address "jay@gmail.com"). The skill body must be self-contained.
• Explicitly declare the key parameters and the final output data types using type hints. Example: `Parameters: param: str; Outputs: output: list[dict]:`
• Include a detailed description of the skill with input and output explanation.
• The skill should not be similar to the existing skills in the skills library.
• The skill must involve multiple processing steps. Simply using the result of an API call without additional logic does not qualify as a valid skill.
• Never call other skills from the skills library or any previously defined skills.
• Do not import any Python packages.
• Avoid a functional style; there's no need to use return.

**3. Update Existing Skills**
Your goal is to ensure the system retains actionable skills that help it behave correctly in the future.
You have three options: **[modify, add, keep]**
• **modify**: revise an existing skill to make it more effective (e.g., improving documents). Only change `content` when necessary, and ensure the resulting skill remains broadly reusable/general-purpose.
• **add**: introduce a new skill only when existing skills cannot support a critical step, in order to improve future performance.
• **keep**: Preserve the skill unchanged when there are no clear issues.
Common actions:
• add a new skill
• update a skill's usage instructions/documentation
• revise a skill's variable/parameter definitions to make it more generalizable
• if a skill is overly complex, refactor it into more modular skills (involving both **modify** and **add**)
• keep a skill unchanged

**4. Requirements for each skill that is modified or added.**
- **Avoid duplication**: If a skills library is provided, do not add new skills that are similar to existing ones—use **keep** or **modify** instead.
- **Exclude non-solution behavior**: Do not include capability exploration, debugging activities, or any failed/incorrect steps.
- **Ensure domain specificity**: The skill must reference domain-specific libraries/APIs, e.g., {api}.
- **Avoid over-wrapping**: Verify the implementation is not merely a thin wrapper around another skill (i.e., not just calling a single underlying skill without meaningful additional logic).
- **Specific-step guided extraction**: Only focus on the specified step in the trajectory when extracting skills.

**5. Good Skill Example**
{example}

**6. Output Format**
You will finish by returning in this JSON format as follows:
```json
[
    {
        "option": "modify",
        "skill": "the modified skill",
        "modified_from": "spotify get all user playlists" # specify the skill name of existing skills that is modified
    },
    {
        "option": "add",
        "skill": "the added skill",
    },
    {
        "option": "keep",
        "skill_name": "the kept skill name",
    }, ...
]
```

Note that your updated skills may not need to cover all the options. You can only use one type of updates or choose to remain all skills unchanged.

**7. CHECKLIST BEFORE FINALIZING**
✓ **Reusability** — Ensure no critical steps are missing, each skill is modular, all parameters are abstract rather than specific.
✓ **Optimality** — Ensure each skill meets the required definition standards.
✓ **Agent-centered** — Add helpful notes in each skill to guide other models in using it correctly.
✓ **Specific-step focus** — Whether the extracted skill includes any content that does not belong to this step?

*Table 8.* Prompt for functional skill extraction based on specific steps.

