# OpenReview forum: "AutoSkills: Automatically Constructing Skill Knowledge Bases for Agents"
_ICML.cc/2026/Conference — Submitted to ICML 2026_

### Official Review · Reviewer_iEYc · 2026-02-21

**Soundness:** 3
**Presentation:** 2
**Significance:** 2
**Originality:** 3
**Overall Recommendation:** 4
**Confidence:** 2

**Summary:**

This paper proposes AutoSkills, an automated framework for constructing a reusable, plug-and-play skill knowledge base for LLM-based agents. The key idea is to distill raw execution trajectories into a hierarchical three-level representation, including planning skills, functional skills, and atomic skills. The framework further incorporates iterative refinement and experience-guided skill expansion to improve skill quality and coverage. Experiments on long-horizon, tool-use benchmarks such as BFCL-v3, AppWorld, and tau^2-Bench demonstrate consistent performance gains when the constructed skill library is transferred to different base agents.

**Compliance With Llm Reviewing Policy:**

Affirmed.

**Final Justification:**

The author’s response addressed most of my concerns. However, given the significant difference in our professional backgrounds, I am unable to assess the novelty of this paper. That said, it appears to me that much effort went into this paper, so I recommend that the AC, SAC, and PC base their decisions primarily on the comments from other reviewers. For now, I am assigning a “weak accept” rating.

**Key Questions For Authors:**

Questions

1.	How robust is the skill library when the backbone model used for extraction is weaker? Would the performance degrade significantly?
2.	How does the skill library scale as the number of tasks or available tools grows significantly? Is there a risk of retrieval noise or context overflow?
3.	Can the authors provide more qualitative examples showing failure cases, especially when the retrieved skills mislead the execution process?

**Limitations:**

No. The author should include a limitation section in the appendix.

**Strengths And Weaknesses:**

Strength

1.	The hierarchical multi-level skill design is well motivated and intuitively structured. The decomposition into planning, functional, and atomic skills provides a clear representation for reusable agent experience.
2.	The automated pipeline (extraction, refinement, expansion) is systematic and reasonably well implemented, showing steady improvements across iterations.
3.	The experimental results are fairly comprehensive, covering multiple benchmarks and different base models, and show consistent performance gains, especially for weaker agents.

Weakness

1.	The overall novelty is moderate. The paper mainly integrates existing ideas (skill abstraction, memory retrieval, iterative refinement) into a unified pipeline, and the conceptual leap over prior skill-based or memory-based agent frameworks is limited.
2.	The reliance on a strong backbone model (GLM-4.6) for skill construction raises questions about scalability. It is unclear how much of the gain comes from the quality of distilled knowledge versus the proposed representation itself.
3.	The skill retrieval and filtering processes involve multiple heuristic design choices (e.g., similarity thresholds, filtering rules), but the sensitivity of performance to these hyperparameters is not sufficiently analyzed.

---

> ### Author Rebuttal · Authors · 2026-03-31
>
> > **Regarding Weakness 1: On novelty: beyond an integration of existing components**
>
> We appreciate the reviewer’s point that AutoSkills draws on several existing threads, including skill extraction, retrieval, and iterative refinement. Our intended contribution is not a claim that each individual ingredient is entirely new in isolation, but that **the specific formulation of reusable agent experience as a pre-built, hierarchical, plug-and-play skill knowledge base is new and practically consequential**, especially in Agentic tool-use settings.
>
> > **Regarding Weakness 2 & Question 1: On reliance on a strong backbone model for skill construction**
>
> We agree that this is an important question. We evaluate AutoSkills on stronger or comparable models, including DeepSeek-V3.2 and GPT-4.1, using either skills extracted by the models themselves or skills constructed by GLM. In all cases, we still observed consistent improvements.
>
> **Due to character limits (5000), we cannot show the ablation experiments in this discussion; please refer to our response to Reviewer fUQY.**
>
> In addition, Table 1 already provides a controlled comparison under the same strong extractor setting. Therefore, these methods share the same advantage of a strong model during the knowledge-construction stage. This suggests that **the key factor is not merely the use of a stronger extractor, but rather the form of experience representation and the experience usage**. Under this controlled setting, AutoSkills still consistently outperforms the alternative methods.
>
> > **Regarding Weakness 3: On heuristic choices in retrieval and filtering**
>
> We appreciate this concern. The current system does include several practical designs, such as similarity thresholds, merging, and filtering rules. We would like to clarify two points.
> - **Parameter configuration**. We did not tune the retrieval mechanism in a benchmark-specific manner, and instead used a single empirical configuration throughout. In addition, LLM-based skill selection further helps ensure the relevance of the retrieved skills.
> - **Quality control**. The skill-iteration mechanisms are mainly introduced to maintain skill quality, rather than being the primary source of the performance gains.
> - **More Evidence from ablation study**. The largest improvements come from the multi-level skill representation and the overall skill library construction pipeline. Iterative refinement and experience-guided expansion provide additional gains. The improvements are consistent across multiple benchmarks and multiple models, suggesting that the method is not the result of narrow tuning to a specific threshold or rule set.
>
> > **Regarding Question 2**
> - **From the perspective of the iteration mechanism**, the size of the skill library is bounded by the maximum diversity of tool usage observed in the training tasks (see Figure 3(d)). Concretely, during iteration, semantically similar skills are merged, while low-quality skills are filtered out. As a result, the library size is expected to converge to an upper bound rather than grow indefinitely.
> - **From the perspective of experience utilization**, we adopt a hybrid retrieval strategy, semantic-similarity-based recall and LLM-based skill selectio, which helps maintain retrieval relevance. In addition, the multi-level skill design makes experience use more efficient and compact, reducing unnecessary context overhead (see Figure 3(f)).
>
> > **Regarding Question 3**
>
> Thank you for the suggestion to analyze failure cases. We find that a common failure mode arises from **an imbalance between following prior experience and exploring the environment**. A representative example is shown below.
>
> **The Skill-Provided Reference Plan**
> ```
> # step 1: Log into the phone app to retrieve Melissa's contact information
> # step 2: Authenticate with Venmo to prepare for making the request
> ... (omitted for brevity)
> ```
>
> **Assistant's first action based on plan**:
> ```
> I'll follow the reference plan to request $28 from Melissa on Venmo.
>
> First, I need to log in to the phone app to retrieve Melissa's contact information.
> ```
> ```
> Error:
> Execution failed. Exception: No API named 'search_contacts' found in the supervisor app.
> ```
> **Over-adherence pattern**:
> 1. The plan says "use phone app to retrieve contact"
> 2. Assistant tried `phone.search_friends`, then `supervisor.search_contacts`
> 3. Neither API exists according to the error
> 4. But the assistant keeps trying variations within the "phone contact lookup" framework
>
> **What should have happened**:
> - Check `show_api_descriptions` for phone app immediately after error
> - Discover correct API names
> - Adapt plan to available APIs

---

> > ### Author Rebuttal · Reviewer_iEYc · 2026-03-31
> >
> > I thank the Authors for their response. I have no further questions, and I will keep my score unchanged.

---

> > > ### Author Response · Authors · 2026-04-01
> > >
> > > We sincerely thank you again for your time, careful reading, and valuable feedback. We greatly appreciate your thoughtful comments and are glad that our rebuttal helped address your concerns. If any remaining questions arise, we would be very happy to further clarify them and provide any additional explanations that may be helpful. Thank you again for your consideration.

---

### Official Review · Reviewer_95hD · 2026-03-10

**Soundness:** 2
**Presentation:** 3
**Significance:** 3
**Originality:** 3
**Overall Recommendation:** 5
**Confidence:** 4

**Summary:**

In this paper, the authors propose AutoSkills framework for constructing a hierarchical skill library from agent trajectories. They do this by extracting planning, functional, and atomic skills using a strong backbone model (GLM 4.6), and then plug this library of skills into weaker agents (for eg. Qwen3) to improve their performance on long-horizon tool-use benchmarks.

**Compliance With Llm Reviewing Policy:**

Affirmed.

**Final Justification:**

I recommend "accept" because the work is practically significant with good transfer results, and the rebuttal addressed all of my concerns.

**Key Questions For Authors:**

1. Can the authors do a fair ablation to separate skill format vs retrieval pipeline? For example: run ExpeL/AWM with the pseudo-plan + step-wise retrieval + LLM selection, or maybe run AutoSkills with only query-based retrieval.
2. Related to previous question. What is the sensitivity to the retrieval threshold and embedding model?
3. Can the authors also report some more metrics like: number of skills (split into planning, functional and atomic sub-types) created during the process, average tokens injected (again split by sub-types), and how often skills are used vs ignored?
4. Do the authors expect the skills to transfer to a different tool suite, basically generalization to another benchmark?

**Limitations:**

No. The paper does not have limitations section. Key limitations:
1. negative transfer observed in experiments (see pt. 1 in Weaknesses)
2. unverified skill correctness
3. no inference-time error recovery + lack of safeguards
are not discussed. I suggest adding explicit limitations discussion, as some of these also affect the applicability of this framework in actual deployment.

**Strengths And Weaknesses:**

## Strengths
- The paper is well-motivated and tackles a real issue in agent learning: agents redo similar tool-use exploration and don’t transfer experience well across tasks or models.
- The planning,functional & atomic split maps cleanly onto common tool-agent failures.
- The transfer results, esp. on Qwen, look promising.

## Weaknesses
1. The paper shows that skill injection can hurt performance for some model-skill combinations (e.g., Qwen3-32B with functional+atomic skills), but provides no analysis of when this occurs or how to predict it.
2. Several equations and notations are misleading.. For e.g.:
-  Equations 9–10 denote skill merging as vector addition (s + δ), but skills are text. I think the authors should use the notation for embeddings in these equations and also specify clearly what happens to those embeddings once they are merged.
- Equation 7 uses set union (∪) for the "modify" operations.
- Equation 5 uses ⊕ without definition.

3. Iteration and skill expansion look risky without strong safeguards. The paper suggests stopping based on test performance, and synthetic task generation can overfit or accidentally get close to test distribution unless you show proper safeguards.
4. Another issue I suspect happening is that there's no inference-time recovery from retrieval errors. E.g. if an incorrect skill is retrieved, agent cannot detect the mismatch or fall back to skill-free reasoning.

---

> ### Author Rebuttal · Authors · 2026-03-31
>
> > **Regarding Weakness 1: Negative transfer for some model-skill combinations**
>
> We agree that this point deserves clearer analysis. Our current results suggest that the effectiveness of skill injection depends on the interaction between model family, skill level, and task match.
>
> A unified interpretation of Figure 3(a) is the following:
>
> - Atomic skills mainly complement the tool schemas that are not fully covered by functional skills. In practice, removing atomic skills often causes the model to overlook certain key schema calls.
> - Functional skills provide stronger behavioral priors, but when the retrieved skill only partially matches the target task, they can also induce over-imitation.
> - Planning skills are often the safest form of transfer, as they provide high-level structure while preserving more flexibility for the model to explore appropriate tool calls.
>
> > **Regarding Weakness 2: Equations and notation are misleading**
>
> Thank you for pointing out this, and we will revise the notation to make the formulation more precise.
> Concretely:
> - In Eq. 5, we will explicitly define ⊕ as partitioned union of three skill subsets.
> - In Eq. 7, we will rewrite this update as an operator over the library state, distinguishing add, modify, and keep explicitly.
> - In Eqs. 9–10, we will clarify this optimization notation in the revised version.
>
> > **Regarding Weakness 3:  Potential Risk overfitting on Iteration and Expansion**
>
> Thank you for raising this important point.
>
> - **From a practical perspective**, we will clarify that early stopping is performed solely based on a held-out validation set, and the test set is used only for final reporting.
> - **From the construction perspective**, the skill expansion process in AutoSkills does not use test sets.
> - **From the quality-control perspective**, AutoSkills already includes safeguards mechanism: General Filter (removing low-quality, overly specialized, or non-transferable skills) and Tool-specific Filter (validating extracted skill content against the tool schema and rejects incompatible tool calls).
>
>
> > **Regarding Weakness 4: No inference-time recovery from retrieval errors**
>
> - **Regarding retrieval errors**, our filtering mechanisms ensure that the main issue is generally not invalid skills, but task–skill mismatch. That is, retrieved skills are typically valid with respect to the tool environment, yet may not always align well with the current task. Our hybrid retrieval strategy partially alleviates this by allowing the model to further select potentially relevant skills, thereby reducing mismatch risk.
> - **Regarding inference-time recovery**, we view this primarily as a harness-engineering issue [1], i.e., whether the execution framework supports an agent loop with failure detection and recovery. AutoSkills is mainly centered on the construction of a plug-and-play skill library rather than on designing a full failure-recovery execution stack. We will make this scope distinction clearer in the limitations discussion. Thank you for advising about this.
>
> > **Regarding Question 1**
>
> Thank you for the helpful suggestion on further analyzing the effects of skill format and retrieval pipeline. We would like to emphasize that, **in experience learning, representation and utilization are inherently coupled: the way experience is stored directly affects how effectively it can be retrieved and reused**. In AutoSkills, the multi-level skill design serves not only as a capability abstraction for agents, but also as a mechanism to improve retrieval recall during skill reuse. In particular, removing pseudo-planning-based recall would likely reduce recall substantially and, consequently, weaken downstream experience utilization.
>
> > **Regarding Question 2**
>
> For the retrieval-related hyperparameters, we did not conduct deliberate, and instead used a single empirical configuration. In practice, semantic-similarity-based retrieval is generally stable, and the subsequent LLM-based skill selection further helps ensure the relevance of the retrieved skills.
>
> > **Regarding Question 3**
>
> We will report additional metrics in the revised version, such as the skill usage ratio, the number of skills and average tokens injected.
>
> > **Regarding Question 4**
>
> Thank you for recognizing the core contribution of our work. **AutoSkills is domain-agnostic, in the sense that it can be applied to new agent environments to automatically construct reusable skill libraries**. We plan to extend this framework to additional benchmarks in future work, further broadening the coverage of the resulting plug-and-play skill library.
>
> > **Regarding Discussing Limitations**
>
> We agree that the paper would benefit from a limitations section, and we will make this explicit in the revision. Specifically, the discussion of Limitation 1 corresponds to our response to Weakness 1, Limitation 2 to Weakness 3, and Limitation 3 to Weakness 4.
>
> [1] Claude Code

---

> > ### Author Rebuttal · Reviewer_95hD · 2026-04-01
> >
> > Thank you for the rebuttal. All my questions and concerns are fully addressed.

---

> > > ### Author Response · Authors · 2026-04-02
> > >
> > > We sincerely thank you again for your time, careful reading, and valuable feedback. We greatly appreciate your thoughtful comments and are glad that our rebuttal helped address your concerns. If any remaining questions arise, we would be very happy to further clarify them and provide any additional explanations that may be helpful. Thank you again for your consideration.

---

### Official Review · Reviewer_fUQY · 2026-03-13

**Soundness:** 2
**Presentation:** 3
**Significance:** 2
**Originality:** 3
**Overall Recommendation:** 5
**Confidence:** 3

**Summary:**

The paper tackles the problem of automatic construction of reusable skill knowledge base using agents that collect and learn from experiences in similar tasks.

The authors showed that the approach outperforms other baselines that learn from agent experiences.

**Compliance With Llm Reviewing Policy:**

Affirmed.

**Final Justification:**

The authors satisfactorily answered all of my previous concerns on the experiments.

**Key Questions For Authors:**

1. could the author provide standard error from the four runs in Table 1.
2. do AutoSkills still provide gains for models that exceed the backbone model in capability? For example, claude or chatgpt models.

**Limitations:**

yes

**Strengths And Weaknesses:**

### Strengths
- The paper is clear and well written
- The experiments are well documented, and the results suggest reasonable gains.
- The analysis on how different skills hierarchy impacts the performance, as well as the detailed case studies in Appendix B, are quite informative

### Weaknesses
- The experiments are mostly done on models less capable than GLM 4.6, which is used as the backbone model for AutoSkills construction. Relative to baselines, there is a natural question on how much does the gain come from prompts generated by a more capable model.
- It is unclear how much does the performance gain come from the simple idea of having agents summarizing its learning in a concise skill format, as opposed to the many architecture designs that the authors introduce (e.g. multi-level skills, skill merge, skill filter etc.). Could the author provide more experimental results demonstrating how the inclusion of each component impacts the performance?
- One claim of the paper is that the skill knowledge base are reusable "across environments". However, the datasets seem too few and homogeneous to support this claim. As number of datasets and necessary skills increase, one might expect proper skill merging, filtering, and skills retrieval to become significantly more difficult.

---

> ### Author Rebuttal · Authors · 2026-03-31
>
> > **Regarding Weakness 1 & Question 2: Evaluation with Stronger Foundation Models**
>
> This is an important question, and we agree it should be addressed explicitly.
>
> - **Evidence from Existing Experiments: The observed gains are not explained solely by stronger-model-generated prompts, because our main table already includes a controlled comparison where the same strong model** (GLM-4.6) is used to extract experience for competing methods. Please refer to Table 1.
>
> - **Additional stronger-model transfer results**: Following your suggestion, we further evaluated AutoSkills on stronger reasoning models, including DeepSeek-V3.2 and GPT-4.1, which are at least comparable to, and in some cases stronger than. The detailed experimental results on Appworld are shown as below.
>
> | Model             | Setting         | Avg@4 | Pass@4 |
> |------------------|-----------------|---------|---------|
> | DeepSeek-v3.2        | no memory           | 61.90   | 84.08   |
> | DeepSeek-v3.2        | self-extract    | **65.48**   | **88.39**   |
> | DeepSeek-v3.2        | glm-extract     | 64.28   | 86.90   |
> | GPT-4.1     | no memory            | 66.37   | 82.74   |
> | GPT-4.1     | self-extract    | **68.60**   | 82.14   |
> | GPT-4.1     | glm-extract     | 66.82   | **84.52**   |
>
> We find that AutoSkills provides consistent performance gains, whether the skills are extracted by these stronger models themselves or constructed using GLM-4.6.
>
> > **Regarding Weakness 2: Unclear Contribution of Individual Components**
>
> We agree that ablation is important for understanding the contribution of each part of the framework. We have now conducted such ablation studies and the results are shown below.
>
> Specifically, Vanilla-Iter1 uses only the multi-level skill design; Vanilla-Iter2 and Vanilla-Iter3 additionally incorporate skill refinement; Expand-Iter1 uses the multi-level skill design together with skill expansion; and Expand-Iter2 and Expand-Iter3 combine the multi-level skill design, skill refinement, and skill expansion. The results suggest that AutoSkills is robust to its experience representation, while skill refinement and expansion can offer further improvements in some cases.
>
> **Model Qwen3-32B**
>
> | Methods       | BFCL-V3 Avg@4 | BFCL-V3 Pass@4 | Appworld Avg@4 | Appworld Pass@4 |
> |-----------|---------------|---------------|----------------|---------------|
> | No Memory     | 53.67         | 73.33          | 27.68         | 47.62          |
> | Vanilla-Iter1 | 59.83         | 78.67          | 30.65         | 54.76          |
> | Vanilla-Iter2 | 62.33         | 79.33          | **35.12**         | 58.93          |
> | Vanilla-Iter3 | 63.67         | 82.00          | 33.33         | 55.95          |
> | Expand-Iter1  | 61.83         | **82.00**          | 32.44         | 54.17          |
> | Expand-Iter2  | 62.33         | 79.33          | 33.48        | **59.52**          |
> | Expand-Iter3  | **64.55**         | 80.00          | 34.23         | 53.57          |
>
> **Model Kimi-K2-Instruct-0905**
>
> | Methods       | BFCL-V3 Avg@4 | BFCL-V3 Pass@4 | Appworld Avg@4 | Appworld Pass@4 |
> |-----------|---------------|---------------|----------------|---------------|
> | No Memory     | 65.17         | 78.00          | 46.88         | 70.24          |
> | Vanilla-Iter1 | 65.50         | 78.80          | 55.80         | 80.95          |
> | Vanilla-Iter2 | 66.83         | 81.33          | 56.40         | **81.55**          |
> | Vanilla-Iter3 | 67.17         | **82.00**          | 53.42         | 79.17          |
> | Expand-Iter1  | 66.00         | 80.00          | 56.40         | 78.57          |
> | Expand-Iter2  | 65.17         | 79.33          | **56.70**         | 77.38          |
> | Expand-Iter3  | **67.33**         | 80.00          | 53.57         | 76.79          |
>
> **Model GLM-4.6**
>
> | Methods       | BFCL-V3 Avg@4 | BFCL-V3 Pass@4 | Appworld Avg@4 | Appworld Pass@4 |
> |-----------|---------------|---------------|----------------|---------------|
> | No Memory     | 76.67         | 83.33          | 60.27         | 83.33          |
> | Vanilla-Iter1 | 78.50         | 85.33          | 62.35         | 83.33          |
> | Vanilla-Iter2 | **79.50**         | **86.00**          | 64.29         | 85.12          |
> | Vanilla-Iter3 | 78.83         | 84.67          | 61.46         | 85.71          |
> | Expand-Iter1  | 78.50         | 85.33          | 64.58         | 83.93          |
> | Expand-Iter2  | 78.83         | 85.33          | 64.88         | 87.50          |
> | Expand-Iter3  | 78.83         | 84.67          | **64.88**         | **88.69**          |
>
>
> > **Regarding Weakness 3: Clarification on the Cross-Environment Generalization Claim**
>
> We appreciate this suggestion and will clarify in the revised version.
> Our core claim is that **AutoSkills enables transfer across foundation agents and generalization to unseen test tasks within each environment**.
>
> **Due to character limits, we cannot show all experiments in this stage, and will present during following discussion**

---

> > ### Author Rebuttal · Reviewer_fUQY · 2026-04-03
> >
> > I appreciate the new experiments, and plan to adjust the score accordingly.

---

> > > ### Author Response · Authors · 2026-04-03
> > >
> > > ### **Gentle Reminder of Adjusting the Score Accordingly**
> > >
> > > **Thank you very much for your positive follow-up and for noting that your concerns have been fully resolved. We truly appreciate your careful reconsideration of the paper.**
> > >
> > > Additionally, we now report the standard error of Avg@4 in Table 1. The results indicate that AutoSkills demonstrates advantages in terms of stability and robustness.
> > >
> > > | Model | Methods | BFCL-V3 Avg@4 | AppWorld Avg@4 | $\tau^2$-Retail | $\tau^2$-Airline | $\tau^2$-Telecom |
> > > |-------|---------|---------------|----------------|-------------|--------------|--------------|
> > > | Qwen3-32B | No Memory* | 53.67 ± 0.45 | 27.68 ± 0.28 | 53.75 ± 0.41 | 38.75 ± 0.47 | 36.25 ± 0.50 |
> > > | Qwen3-32B | A-Mem* | 53.67 ± 0.43 | 26.79 ± 0.26 | 53.12 ± 0.42 | 38.75 ± 0.47 | 38.12 ± 0.52 |
> > > | Qwen3-32B | AWM* | 55.67 ± 0.44 | 30.80 ± 0.25 | 55.00 ± 0.39 | 40.00 ± 0.47 | 38.12 ± 0.52 |
> > > | Qwen3-32B | AWM‡ | 56.67 ± 0.42 | 34.45 ± 0.25 | 57.50 ± 0.39 | 41.25 ± 0.46 | 40.62 ± 0.50 |
> > > | Qwen3-32B | ExpeL* | 57.33 ± 0.42 | 32.87 ± 0.22 | 56.25 ± 0.38 | 42.50 ± 0.45 | 39.38 ± 0.48 |
> > > | Qwen3-32B | ExpeL† | 59.33 ± 0.40 | 32.94 ± 0.21 | 58.12 ± 0.37 | 43.75 ± 0.45 | 41.25 ± 0.46 |
> > > | Qwen3-32B | AutoSkills‡ | 63.67 ± 0.39 | 35.12 ± 0.21 | 66.87 ± 0.36 | 47.50 ± 0.44 | 43.75 ± 0.46 |
> > > | Kimi-K2-Instruct-0905 | No Memory* | 65.17 ± 0.42 | 46.88 ± 0.13 | 75.62 ± 0.27 | 51.25 ± 0.47 | 78.12 ± 0.16 |
> > > | Kimi-K2-Instruct-0905 | A-Mem* | 65.17 ± 0.40 | 46.58 ± 0.13 | 76.25 ± 0.24 | 52.50 ± 0.44 | 76.87 ± 0.17 |
> > > | Kimi-K2-Instruct-0905 | AWM* | 65.33 ± 0.42 | 49.70 ± 0.12 | 76.25 ± 0.26 | 53.75 ± 0.46 | 77.50 ± 0.18 |
> > > | Kimi-K2-Instruct-0905 | AWM‡ | 64.67 ± 0.41 | 50.60 ± 0.12 | 76.25 ± 0.26 | 53.75 ± 0.45 | 77.50 ± 0.18 |
> > > | Kimi-K2-Instruct-0905 | ExpeL* | 66.33 ± 0.43 | 52.53 ± 0.12 | 77.50 ± 0.21 | 55.50 ± 0.42 | 78.75 ± 0.16 |
> > > | Kimi-K2-Instruct-0905 | ExpeL† | 66.00 ± 0.41 | 52.98 ± 0.10 | 77.50 ± 0.20 | 56.25 ± 0.42 | 79.37 ± 0.16 |
> > > | Kimi-K2-Instruct-0905 | AutoSkills† | 66.83 ± 0.39 | 56.40 ± 0.09 | 78.12 ± 0.16 | 58.75 ± 0.41 | 82.50 ± 0.16 |
> > > | GLM-4.6 | No Memory* | 76.67 ± 0.37 | 60.27 ± 0.21 | 76.25 ± 0.30 | 70.00 ± 0.31 | 70.63 ± 0.30 |
> > > | GLM-4.6 | A-Mem* | 76.50 ± 0.36 | 60.57 ± 0.21 | 76.88 ± 0.26 | 70.00 ± 0.32 | 68.75 ± 0.29 |
> > > | GLM-4.6 | AWM* | 77.17 ± 0.38 | 62.20 ± 0.22 | 77.50 ± 0.27 | 71.25 ± 0.36 | 70.63 ± 0.28 |
> > > | GLM-4.6 | ExpeL* | 78.83 ± 0.35 | 64.14 ± 0.16 | 77.50 ± 0.23 | 72.50 ± 0.34 | 71.25 ± 0.26 |
> > > | GLM-4.6 | AutoSkills* | 79.50 ± 0.35 | 64.88 ± 0.14 | 82.50 ± 0.16 | 76.25 ± 0.31 | 71.88 ± 0.22 |
> > >
> > > ***We hope these additional results further support the paper’s claims. If you feel that the concerns have now been adequately addressed, we would be very grateful if your final overall assessment could reflect this updated evaluation.***

---

### Official Review · Reviewer_7ouP · 2026-03-13

**Soundness:** 3
**Presentation:** 3
**Significance:** 3
**Originality:** 3
**Overall Recommendation:** 4
**Confidence:** 3

**Summary:**

This paper introduces AutoSkills, a framework for automatically constructing and maintaining a plug-and-play skill knowledge base for LLM-based agents. Specifically, this paper distills raw interaction trajectories into a three-level hierarchy of reusable skills, combines automatic skill evolution, performs iterative skill refinement to improve skill quality based on execution feedback, and enables exploratory skill expansion, allowing the agent to generate and validate new candidate skills beyond those directly observed in trajectories.

**Compliance With Llm Reviewing Policy:**

Affirmed.

**Ethical Review Concerns:**

yes

**Key Questions For Authors:**

1. The framework relies on repeated LLM calls for skill extraction, refinement, and expansion. Could the authors provide additional discussion on the scalability of this process, including:

    (1) the computational overhead of constructing and maintaining the skill library,

    (2) the typical size and growth of the skill repository

    (3) how the retrieval mechanism scales as the number of skills increases?

2. As the skill library grows through iterative refinement and exploratory expansion, how does the system handle potential redundancy or conflicts between skills? For example, multiple skills may represent similar behaviors with slightly different abstractions. Does the framework include mechanisms for skill deduplication, merging, or pruning to maintain a compact and consistent knowledge base?

**Limitations:**

It would be helpful for the authors to discuss possible failure modes (e.g., propagation of incorrect or unsafe skills during iterative refinement), as well as risks related to deploying automatically generated skills across different agents or environments without sufficient verification.

**Strengths And Weaknesses:**

### Strengths:
1. The paper addresses an important problem in LLM agents: enabling systematic accumulation and reuse of experience rather than repeatedly solving tasks from scratch.
2. The framework has practical implications for improving efficiency and generalization in long-horizon agent tasks, as it constructs a reusable and transferable skill knowledge base that can be plugged into different agents via a lightweight retrieval module.
3. The paper is generally clear and easy to follow.



### Weaknesses:
1. The framework combines Multi-Level Skills Design, Iterative Skills Refinement, and Exploratory Skills Expansion, but the individual contribution of each component is somewhat unclear. The authors could provide more detailed ablation studies isolating these components (e.g., removing refinement or expansion) to better understand which parts contribute most to the observed performance gains.
2. Since a key claim of the paper is that the skill knowledge base enables reusable experience across agents and tasks, it would be helpful to evaluate whether skills extracted in one environment or task domain can meaningfully improve performance in different domains or unseen tasks.

---

> ### Author Rebuttal · Authors · 2026-03-31
>
> > **Regarding Weakness 1: Unclear Contribution of Individual Components**
>
>
> Thank you for your suggestion regarding the ablation study of the AutoSkills component. We acknowledge that the original submission did not include a sufficiently complete analysis, partly due to time constraints during the earlier development stage. In response, we have now conducted more comprehensive set of ablation experiments on the AppWorld and BFCL-v3 benchmarks using Qwen3-32B, K2, and GLM-4.6.
>
>
> Specifically, Vanilla-Iter1 uses only the multi-level skill design; Vanilla-Iter2 and Vanilla-Iter3 additionally incorporate skill refinement; Expand-Iter1 uses the multi-level skill design together with skill expansion; and Expand-Iter2 and Expand-Iter3 combine the multi-level skill design, skill refinement, and skill expansion. These results suggest that AutoSkills is robust to its underlying experience representation, while iterative refinement and skill expansion can offer further improvements depending on the model
> and the particular combination of components.
>
>
> **Due to character limits (5000), we cannot show the ablation experiments in this discussion; please refer to our response to Reviewer fUQY.**
>
>
> Please note that we did not perform the skill refinement and skill expansion component ablations on tau2-bench. This is because tau2-bench is a user-interactive benchmark whose tool schemas are relatively simple in both number and dependency structure, and its training set already covers many task patterns directly. More broadly, for user-centric benchmarks of this type (e.g., dialogue benchmarks such as LocMO), **it remains an open question whether experience learning centered around tool-schema-based skills is the most appropriate formulation**. Therefore, we believe that component studies on skill iteration and skill expansion are less suitable for tau2-bench, and we did not include them in our ablation experiments.
>
>
> > **Regarding Weakness 2: Evaluating Cross-Domain Reusability**
>
> We agree with your suggestion that true cross-environment generalization would expand the boundary of our framework. The skills we define in this paper are centered in domain-specific tool schemas.
> We will revise the paper to make this issue more explicit. Our core claim is that, **within a given tool environment, AutoSkills enables strong transfer across agents and tasks**.
> Thank you for  suggesting us to better clarify the scope of our claims.
>
> > **Regarding Question 1: Scalability (Computational Overhead, Library Size, Retrieval Scaling)**
>
> Thank you for these crucial questions on the practical aspects of AutoSkills.
> - **Analysis of Computational Overhead**: We would like to clarify that our target setting is offline, potentially costly construction of a high-quality skill library, which can then be reused efficiently during online deployment. Second, we find that using the skill library results in an average of 22.18 LLM calls per trajectory.
> - **Analysis of Library Size**: We would like to clarify that the skill refinement in AutoSkills is specifically designed to avoid uncontrolled growth of the skill library under a fixed tool environment.
> Additionally, we also report how the Skill Library Size evolves over more iterations.
> **Due to character limits (5000), we cannot show all experiments at this stage, and will present during following discussion**
>
> - **Analysis of Computational Retrieval Mechanism**: Our retrieval module adopts a hybrid strategy based on standard FAISS indexing and LLM-guided selection. This provides both efficient retrieval and flexible matching based on functional similarity, so retrieval is not a bottleneck in practice.
>
> > **Regarding Question 2: Handling Redundancy or Conflicts**
>
> Regarding maintenance of the skill library, Figure 2 shows that skill refinement includes four key steps: semantic-similarity-based clustering, merging functionally similar skills, decomposing overly complex skills, and filtering. These mechanisms jointly help preserve skill quality while controlling unnecessary growth of the library. Furthermore, each skill can undergo one of three update operations: *add*, *modify*, or *keep*. In particular, imperfect skills are refined by revising the original skill rather than naively introducing new skill.
>
> > **Regarding Discussing Limitations**
>
> Thank you for pointing out the need for a clearer discussion of the scope and limitations of our work. In the revised version, we will discuss cross-environment transfer and the extension of AutoSkills to agent settings without tool use.

---

> > ### Author Rebuttal · Reviewer_7ouP · 2026-04-06
> >
> > My concerns have been addressed.

---

> > > ### Author Response · Authors · 2026-04-06
> > >
> > > We sincerely thank you again for your time, careful reading, and valuable feedback. We greatly appreciate your thoughtful comments and are glad that our rebuttal helped address your concerns. If any remaining questions arise, we would be very happy to further clarify them and provide any additional explanations that may be helpful. Thank you again for your consideration.

---

### Decision · Program_Chairs · 2026-04-30

**Decision:**

Reject

**Comment:**

The paper describes a system for automatically learning skills for LLM-based agents. All reviewers agreed with the recommendation of accepting the paper.

A major weakness that wasn't addressed during the rebuttal is the lack of scholarship: the paper ignores a large body of literature on skill learning in sequential decision-making. The paper cites a recent piece by Rich Sutton, "Welcome to the Era of Experience", but fails to cite Sutton, Precup & Singh's 1999 seminal work on the options framework, which essentially describes the use of skills in reinforcement learning. The current paper would benefit from a careful review of skill learning in machine learning. The oldest paper cited in this submission is from 2022. To me, this is a serious weakness that should be addressed prior to publication.

I understand the skills presented in this work are not the same as RL skills, but it is still important to make this distinction clear. However, if we step back and look at LLM-based and RL skills, we will note they aren't different. In RL, a skill is a program the agent can call that instructs it for a number of steps (temporal abstraction). In the LLM-agent sense, it is a prompt that works as a natural-language program, also offering... temporal abstraction.

Currently, the paper does not "talk to prior work", and this is a major issue in terms of scholarship. This is why I recommend that the paper be rejected in its current form, with strong encouragement to resubmit to another flagship conference.